# EEG-Based Emotion Recognition via Prototype-Guided Disambiguation and Noise Augmentation in Partial-Label Learning

## Abstract

EEG-based emotion recognition offers an objective method for diagnosing emotion-related health issues, but the inherent complexity of emotions often leads to annotation errors and noisy labels. To simulate this labeling process in emotion recognition, we propose a semantic-based candidate label generation method leveraging the GloVe vectors, which considers the semantic relationships between emotions. Under the Partial Label Learning (PLL) scenario, we introduce a novel model called PGNA-PL (Prototype-Guided Noise-Augmented Partial Label Learning). This model learns inter-class relationships of emotions using prototypes, and uses a self-distillation mechanism to iteratively guide the classifier's disambiguation process. To address the low signal-to-noise ratio (SNR) of EEG, we introduce a noise augmentation strategy inspired by the mixup method, incorporating controllable noise to enhance model robustness. Experiments on three public datasets (SEED, SEED-IV, SEED-V) show that our approach achieves state-of-the-art performance, surpassing existing PLL baselines across different candidate label generation modes. Our method effectively disambiguates complex emotions and shows promising results in assisting in the recognition of fear-related disorders.

## 1 Introduction

The interplay between human emotions and overall health, both psychological and physiological, is profound, with evidence showing a strong linkage between negative emotional states and conditions like phobias (Apicella et al., 2024). Physiological signals, due to their difficult-to-disguise nature, are suitable for the objective assessment of emotional responses (Liu et al., 2024; Li et al., 2022b; Shen et al., 2022). Research has demonstrated the viability of extracting emotion-related features from various physiological signals, including electroencephalogram (EEG) (Wang et al., 2024a), eye movements (Zheng et al., 2018), and peripheral physiological signals (Koelstra et al., 2011). Among these, EEG due to its precision and high temporal resolution, has already been extensively studied (Zhang et al., 2022b; Liu et al., 2024; Ding et al., 2022).

To obtain high-quality EEG-based emotion datasets, auditory and visual stimuli are typically employed to induce emotional responses in participants, with EEG signals recorded through specialized equipment. While the labeling of EEG signals generally relies on expert annotations (Zheng & Lu, 2015) or participants' self-reports (Koelstra et al., 2011), the inherent complexity of emotions often leads to ambiguities, resulting in annotation errors (Jiang et al., 2024). This challenge is nearly unavoidable. For example, as shown in Plutchik's wheel of emotions (Plutchik, 1980), the emotion "remorse" encompasses both "disgust" and "sadness". When participants experience such complex, overlapping emotions, the labeling process becomes difficult. This is especially prevalent when dealing with rich emotional expressions in audiovisual stimuli.

Partial label learning (PLL) (Cour et al., 2011) offers a more flexible approach by allowing multiple potential labels to be assigned simultaneously, with the assumption that only one label is correct. In this context, the labeled tags are referred to as candidate labels, while the unlabeled ones are considered non-candidate labels (Wen et al., 2021). Under this assumption, the goal during training is to disambiguate the multiple candidate labels, enabling the model to learn how to correctly identify

the true label from the ambiguous annotations. The introduction of this method can significantly mitigate the impact of labeling errors in EEG-based emotion recognition tasks Zhang & Etemad (2023) .

However, in emotion classification tasks, different emotions often share latent relationships, making the traditional PLL assumption (Wang et al., 2022; Xia et al., 2023) of uniformly distributed candidate labels unsuitable. Furthermore, other methods like Zhang & Etemad (2023) overlook the semantic relations between emotions. To address this challenge for the first time, we propose a novel semantic-based candidate label generation method that explicitly incorporates semantic relationships between emotions. By leveraging the GloVe vectors (Pennington et al., 2014) (detailed in Appendix A.2.1), which is extensively used in natural language processing (NLP) to capture contextual word relationships, we obtain semantic vectors for various emotions. These vectors represent emotions in a multi-dimensional space, reflecting their contextual relationships based on large text corpora. We then use cosine similarity to calculate the distances between these emotion vectors, which are instrumental in computing the probability of each emotion being selected as a candidate label. This method helps in quantitatively assessing the similarity or dissimilarity between emotional states, thereby enabling more accurate candidate label generation.

PLL algorithms have been extensively studied in computer vision (Tian et al., 2023), yet their potential in EEG-based emotion recognition has not been fully exploited. A key challenge is understanding and utilizing the inter-class relationships of emotions to aid in label disambiguation. Additionally, the inherently low SNR of EEG signals (Ye et al., 2022; Huang et al., 2024; Li et al., 2024b), exacerbated by issues like electrode displacement due to head movements (Wang et al., 2024b), further complicates the analysis.

To solve these challenges, we introduce the PGNA-PL model (Prototype-Guided Noise-Augmented Partial Label Learning). This model stabilizes emotion relationships using prototypes and employs a self-distillation approach (Zhang et al., 2019; Li et al., 2024c) to guide the classifier's disambiguation process among emotions. To mitigate low SNR issues, we incorporate a controllable noise augmentation technique inspired by the mixup method (Zhang et al., 2018). This method mixes features from a small number of other samples to create perturbations closer to the distribution of EEG signals without altering the label, thereby enhancing the robustness of the model.

We evaluate our method on three publicly available datasets: SEED (Zheng & Lu, 2015), SEED-IV (Zheng et al., 2018), and SEED-V (Liu et al., 2021), under two different candidate label generation strategies, including our proposed Semantic Distribution and the real-world experimental framework Zhang & Etemad (2023) referred to as Russell Distribution in this paper for simplicity (detailed in Appendix A.2.2). PGNA-PL achieves state-of-the-art (SOTA) performance across all datasets. Analysis of the confusion matrix suggests that our method shows potential advantages in assisting in the recognition of fear-related disorders..

By enhancing the accuracy and robustness of emotion recognition from EEG signals and effectively addressing labeling errors through partial label learning, our proposed methods have the potential to transform applications in mental health diagnostics, adaptive user interfaces, and affective computing, providing more reliable and objective assessments of emotional states.

Our contributions are threefold:

1. We introduce a novel semantic-based candidate label generation method for PLL that, for the first time, incorporates the semantic relationships of emotions. This method not only addresses the limitations of previous approaches but also establishes a new evaluation framework applicable to emotion recognition tasks.

2. We propose the PGNA-PL model, an advanced approach utilizing prototypes to decode emotional relationships between classes and self-distillation for enhanced classification accuracy. This model is further augmented with a unique noise augmentation technique, improving its robustness.

3. Through comprehensive evaluations on three public datasets, the PGNA-PL model demonstrates superior performance over existing methods. Notably, the model shows exceptional efficacy in detecting fear emotion, which could have significant implications for advancements in mental health assessments.

## 2 METHODS

### 2.1 OVERVIEW

To more accurately reflect the inter-class relationships of emotions in the candidate labels used for evaluation, this section first discusses how to generate candidate labels with an understanding of emotional semantics. Subsequently, we introduce a novel EEG-based emotion recognition model PGNA-PL that employs prototypical representations to construct inter-emotional category relationships and enhances the classifier's emotional classification capabilities through self-distillation. Additionally, we propose a tailored noise augmentation strategy for EEG signals to enhance the model's robustness. The pseudo-code of PGNA-PL is shown in Algorithm 1.

### 2.2 GENERATING EMOTIONALLY SEMANTIC CANDIDATE LABELS

Human understanding of emotions is inherently based on their semantic relationships, which can be captured from large textual corpora. To model these semantics, we represent each emotion label using GloVe vectors (Pennington et al., 2014). Let $v_i$ denote the GloVe vector for emotion category $e_i$, where $i$ indexes the emotion vocabulary.

To quantify the semantic similarity between two emotions $e_i$ and $e_j$, we compute the cosine similarity between their corresponding vectors:

$$s_{ij} = \frac{v_i \cdot v_j}{\|v_i\| \, \|v_j\|} \tag{1}$$

This similarity score $s_{ij}$ reflects how closely related the two emotions are in semantic space.

In generating candidate labels for emotion classification, we leverage these semantic similarities. For a given primary emotion category $e_i$, we use the cosine similarities $s_{ij}$ to probabilistically assign additional emotion labels. Inspired by (Zhang & Etemad, 2023), for each emotion category $e_j$, we generate a binary label $\hat{Y}[j]$ by sampling from a Bernoulli distribution parameterized by $s_{ij}$:

$$\hat{Y}[j] \sim \text{Bernoulli}(s_{ij}) \tag{2}$$

Here, $\hat{Y}$ is the candidate label vector for the current sample, and $\hat{Y}[j]$ indicates whether emotion $e_j$ is included as a candidate label.

After generating the binary labels, we normalize $\hat{Y}$ to form a probability distribution, where k is the number of emotion categories:

$$\hat{Y} = \frac{\hat{Y}}{\sum_{j=1}^{k} \hat{Y}[j]} \tag{3}$$

This normalization ensures that the candidate labels sum to one, facilitating their use in probabilistic models.

Equations (2) and (3) are applied sequentially to the emotion labels of each sample. As a result, samples with the same true label may be assigned different candidate labels.

By incorporating semantic similarities between emotions, this method better simulates the occurrence of label errors in partial label scenarios, improving the representation of the labeling process.

### 2.3 PROTOTYPE-GUIDED NOISE-AUGMENTED PARTIAL LABEL LEARNING MODEL

We utilize the differential entropy (DE) features of EEG signals (Duan et al., 2013) as input to our model. Let $X$ denote the input feature space, and $Y = \{1, 2, \ldots, k\}$ represent the set of possible emotion labels. Our dataset $D$ consists of $n$ samples and is defined as $D = \{(x_i, y_i)\}_{i=1}^{n}$, where $x_i \in X$ is an EEG sample, and $y_i \in Y$ is the corresponding one-hot encoded candidate label set generated as described in Section 2.2.

As illustrated in Figure 1, the PGNA-PL model comprises an encoder $E$ and a classifier $C$, parameterized by $\theta$ and $\delta$, respectively. The encoder $E$ extracts high-level features from the input data, and the classifier $C$ assigns these features to specific emotion categories. The functions of $E$ and $C$ are denoted by $f_\theta$ and $F_\delta$.

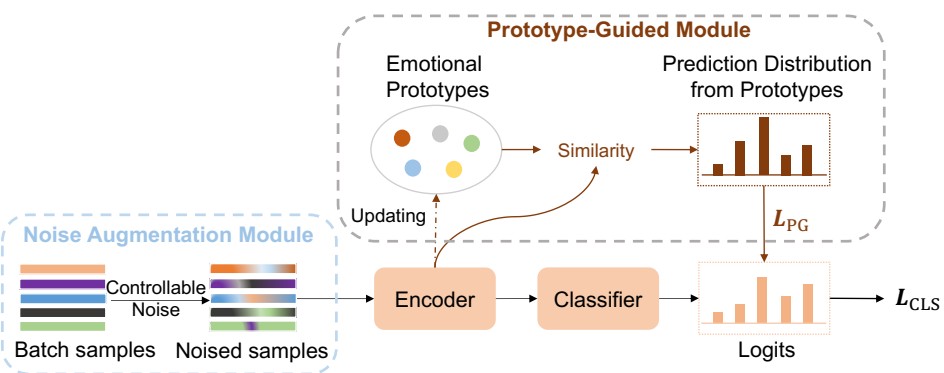

Figure 1: The PGNA-PL model, use the Prototype-Guided Module and the Noise Augmentation Module to improve the model's disambiguation capability and robustness, respectively.

When an input sample $x_i$ is processed through the encoder, it generates a high-level feature representation $p_i$:

$$p_i = f_\theta(x_i). \tag{4}$$

This high-level feature is then fed into the classifier to obtain the predicted logits $\hat{y}_i$:

$$\hat{y}_i = F_\delta(p_i). \tag{5}$$

Under the supervision of the candidate emotion labels, $\hat{y}_i$ represents the probability distribution of the classifier's predictions across different emotion categories.

To enhance the model's emotion recognition capabilities, we introduce a prototype-guided module and a noise augmentation module, resulting in the proposed Prototype-Guided Noise-Augmented Partial Label Learning (PGNA-PL) model.

### 2.3.1 PROTOTYPE-GUIDED MODULE

In PLL, the classifier's predictions may be unreliable due to label ambiguity. To address this issue, we propose a prototype-guided approach that constructs emotion prototypes, with the goal of capturing inter-class relationships. Unlike the PiCO method (Wang et al., 2022), which updates candidate labels using prototypes and may suffer from instability due to frequent changes in the training objective, our method avoids such instability by maintaining a fixed learning objective while still leveraging prototypes to guide the classifier's learning process.

We define emotion prototypes $\mu_k \in \mathbb{R}^d$, where $d$ is the feature dimensionality of each emotion category, and $k$ is the number of emotion categories. Initially, each prototype is set to the zero vector of dimension $d$. During training, we select the emotion category corresponding to the highest predicted probability in the logits $\hat{y}_i$, and update its associated prototype using a moving average mechanism in (Wang et al., 2022). This allows the prototype to evolve over time, reflecting the input data distribution. Formally, the prototype update rule is given by:

$$\mu_k = \text{Normalize}\left(\xi\mu_k + (1-\xi)p_i\right), \quad \text{if} \quad k = \arg\max_k(\hat{y}_i) \tag{6}$$

Here, $\xi$ is a hyperparameter controlling the rate of prototype update, typically set close to 1 to ensure small, incremental changes. The feature vector $p_i$ is the output of the encoder for the $i$-th training sample, and normalization ensures that the prototype vector remains within a consistent scale.

Over time, these prototypes represent the various emotion categories, and the distances between them capture the relationships between different emotions. As a result, the classifier can more easily distinguish between similar emotions based on the evolving prototypes.

Once the prototypes have been updated, we use them to compute a probability distribution $\hat{y}_\mu$ that represents the likelihood of an input belonging to each emotion category. This is done by calculating the similarity between the feature vector $p_i$ and each of the prototypes $\mu_k$. Specifically, we compute the similarity between the feature vector and each prototype:

$$\hat{y}_\mu = p_i \cdot \mu_k^T \tag{7}$$

Here, $p_i \cdot \mu_k^T$ denotes the dot product between the feature vector $p_i$ and the transpose of the prototype.

To guide the classifier's learning, we employ self-distillation. This involves aligning the classifier's logits $\hat{y}_i$ with the probability distribution $\hat{y}_\mu$ derived from the prototypes. Specifically, we minimize the Kullback-Leibler (KL) divergence between these two distributions, which helps the classifier learn the inter-class relationships encoded in the prototypes. The Prototype Guidance loss $L_{PG}$ is computed as:

$$L_{PG} = \frac{1}{m} \sum_{i=1}^{m} \text{KL}\left(\text{softmax}\left(\frac{\hat{y}_i}{\tau}\right), \text{softmax}\left(\frac{\hat{y}_\mu}{\tau}\right)\right) \tag{8}$$

Here, $\tau$ is a temperature parameter that softens the probability distributions, and $m$ represents the batch size. By minimizing this loss, the classifier is guided to produce output distributions that are consistent with the inter-class relationships learned from the prototypes.

In summary, the Prototype-Guided Module introduces emotion prototypes that evolve over time, helping the classifier to better understand and differentiate between emotion categories. Through self-distillation, the classifier learns to align its predictions with the relationships captured by the prototypes, thereby improving its ability to handle label ambiguity and inter-class similarity. This approach avoids the instability associated with directly updating candidate labels and enhances the model's overall performance in emotion classification tasks.

### 2.3.2 Noise Augmentation Module

In EEG signal processing, one of the key challenges is the low signal-to-noise ratio (SNR), which can significantly hinder the model's performance. To address this, we propose a noise augmentation method inspired by consistency regularization (Yan & Li, 2021; Wu et al., 2022), which adds controlled noise perturbations to the input EEG signals without changing their candidate labels. The goal is to force the model to produce consistent outputs despite the noise.

However, due to the continuous and time-series nature of EEG signals, traditional data augmentation techniques, such as cropping or scaling, are not applicable (Li et al., 2022a). Moreover, generating new signals using complex models like Generative Adversarial Networks (GANs) would dramatically increase the model's complexity. Instead, we draw inspiration from the mixup technique (Zhang et al., 2018), which is commonly used for data augmentation. In mixup, two samples from the same batch are randomly mixed, and the corresponding labels are also interpolated. This helps the model generalize better. However, unlike the traditional mixup, our approach focuses on adding noise by blending two EEG signals, thus maintaining the structure of the signal while introducing perturbations that simulate noise.

To begin, we sample a mixing coefficient $\lambda$ from a Beta distribution, as described in Equation equation 9. The Beta distribution is parameterized by a hyperparameter $\beta_p$, which determines the range of $\lambda$. For simplicity, we use equal values for both parameters for the Beta distribution. If $\beta_p$ is small, $\lambda$ will be close to 0 or 1, leading to a higher contribution from one of the signals. Conversely, larger values of $\beta_p$ make $\lambda$ closer to 0.5, promoting a more balanced mix of the two signals.

$$\lambda = \text{Beta}(\beta_p, \beta_p) \tag{9}$$

Next, we apply a scaling factor $\sigma$ to constrain the mixing coefficient $\lambda$. This ensures that the mixed sample retains more of the original signal's features. Specifically, $\sigma$ is sampled from the range $(0.5, 1]$ to ensure that the perturbations do not distort the original sample too much. The final mixing coefficient $\lambda'$ is computed as:

$$\lambda' = \sigma + (1 - \sigma) \cdot \lambda \tag{10}$$

With the mixing coefficient $\lambda'$ determined, we mix two EEG signals, $x_i$ and $x_j$, sampled from the same batch. The new noise-augmented signal is given by:

$$x_i' = \lambda' \cdot x_i + (1 - \lambda') \cdot x_j \tag{11}$$

This mixed signal $x_i'$ is then used as the input to the model. The key advantage of this approach is that it allows us to simulate noise without altering the candidate labels, maintaining the integrity of the original classification task while adding perturbations that improve generalization.

After conducting a coarse grid search over different values of $\lambda'$, we found that values in the range of $[0.8, 0.9]$ yield optimal performance. This indicates that maintaining a higher proportion of the original sample helps preserve its features, making the noise perturbations more controlled and stabilizing the training process.

While similar interpolation methods have been explored in the context of data augmentation, such as in PaPi (Xia et al., 2023), our approach differs in its primary goal: adding noise rather than augmenting data purely for label interpolation. In PaPi, candidate labels are updated using the classifier's predictions on new interpolated samples, but the focus is on generating diverse data rather than introducing noise. In contrast, our method ensures that the candidate labels remain unchanged, while the noise addition process is controlled via the scaling of $\lambda$, as described in Equation equation 10. This helps maintain the utility of the original labels for disambiguation.

In summary, the noise augmentation module addresses the low SNR in EEG signals by adding controlled noise perturbations. Using a mixup-based method, we blend two EEG signals from the same batch, adjusting the mixing ratio through a scaling factor that ensures the original sample is preserved while adding noise. This method stabilizes training by forcing the model to be consistent in the presence of noise, without altering the candidate labels, thereby improving robustness.

### 2.3.3 CLASSIFICATION LOSS AND OVERALL LOSS

To train the model with partial labels, we use the naive loss function from DNPL (Seo & Huh, 2021), which keeps the candidate labels fixed. The classification loss $L_{CLS}$ is computed as follows:

$$L_{CLS} = -\frac{1}{m} \sum_{i=1}^{m} \log \left( \upsilon \left( \sum_{\text{class}=1}^{k} \text{Softmax}(\hat{y}_i) \cdot y_i \right) \right) \tag{12}$$

Here, $\hat{y}_i$ represents the logits for the $i$-th sample, and $y_i$ denotes the corresponding candidate label. First, we apply the softmax function to the logits to obtain the probability distribution over the $k$ possible emotion categories. Then, the probabilities corresponding to the candidate labels are summed, yielding the total probability of correct recognition for each sample. This value is passed through the operation $\upsilon$, which clamps the result between 0 and 1 to stabilize the output. The logarithmic function $\log$ is then used to compute the loss. The batch size $m$ is used to average the loss across the batch.

The overall loss $L_{overall}$ integrates the emotion classification loss $L_{CLS}$ and the Prototype Guidance loss $L_{PG}$ through the balancing hyperparameters $\alpha$, as follows:

$$L_{overall} = L_{CLS} + \alpha \cdot L_{PG} \tag{13}$$

### 2.3.4 ARCHITECTURE DETAILS

The architecture of the encoder and classifier is derived from (Zhang et al., 2022a; Zhang & Etemad, 2023), as they have been demonstrated to be superior in the task of EEG emotion recognition. The encoder consists of two CNN modules; each includes a 1-D convolutional layer, batch normalization, and a LeakyReLU activation function. The output is then transformed to the desired dimensions $d$ through a flattening operation and a linear layer. The classifier comprises a single linear layer designed for emotion recognition. Further details can be found in Appendix A.3.

## 3 EXPERIMENTS

### 3.1 EXPERIMENTAL SETUP

Our experiments are conducted on three publicly accessible EEG emotion datasets: SEED, SEED-IV, and SEED-V, using the DE features provided by the official sources. The datasets include 3 (positive, negative, neutral), 4 (happy, sad, neutral, fear), and 5 (happy, sad, disgust, neutral, fear) emotional categories, respectively. Each dataset consists of three sessions, which means that each subject participates in three separate full experiments under varying visual stimuli. Detailed descriptions of the datasets can be found in Appendix A.4.

---

**Algorithm 1** PGNA-PL method

---

1: **Input:** iteration $T$, dataset $D = \{(x_i, y_i)\}_{i=1}^n$
2: **Output:** optimized PGNA-PL model
3: **The Training Phase:**
4: Randomly initialize $\theta$ and $\delta$. Initialize $\mu$ as all-zero vectors.
5: **for** $t = 1$ **to** $T$ **do**
6:     Sample a mini-batch of training data.
7:     Obtain $p$ and the prediction logits $\hat{y}_i$ for the batch data by Equation (4) and (5).
8:     Add noise to the batch data by Equation (9-11).         ▷ Noise augmentation
9:     Obtain $\hat{y}_\mu$ by Equation (7) for the batch data.
10:    Update $\mu$ by Equation (6) using the batch data.        ▷ Update the prototype
11:    Compute $L_{PG}$ by Equation (8).                ▷ Prototype guidance
12:    Compute $L_{CLS}$ by Equation (12).             ▷ Emotion recognition
13:    Compute $L_{overall}$ by Equation (13).        ▷ Compute the overall loss
14:    Update $\theta$ and $\delta$ by backpropagation.
15: **end for**
16: Return $\theta$ and $\delta$.

---

To ensure robust evaluation across different datasets, we account for their distinct characteristics by splitting them into train, validation, and test sets. Specifically, in the SEED dataset, each session consists of 15 emotion-specific EEG trials. We follow traditional protocols by using the first nine trials for the training set and the remaining six for the test set. The final results are reported as the average performance across the test sets of the three sessions. The SEED-IV dataset, with 24 trials per session, employs a unique approach due to the random order of trials. We evenly divide these trials into three parts by category, taking one part from each session (8 trials) to form a fold in a three-fold cross-validation process, with a 36:18 split of training set to test set trials. The SEED-V dataset adopts a similar design to SEED-IV for cross-validation without the need for sample balancing, since five videos of different emotions always appear sequentially, leading to a 30:15 ratio of training set to test set trials. We randomly select 20% of the training set to serve as the validation dataset, with the remaining data used for training the model. Our evaluation metrics include accuracy (standard deviation), macro F1, and micro F1 to ensure balanced assessment under multi-class conditions. Details on the settings of the hyperparameters can be found in Appendix A.5.

Notably, semantic-based candidate labels for the datasets are standardized by mapping original emotion categories to adjectives for uniformity, such as mapping SEED-V categories to (happy, sad, disgusted, neutral, fearful). In the SEED dataset, where videos primarily induce happy or sad emotions, we refine these categories to "happy" and "sad" for clearer semantic resonance, retrieving emotion semantic vectors from GloVe based on these terms. Additionally, we evaluate all methods using the Russell Distribution to validate model generalizability.

## 3.2 RESULTS

### 3.2.1 COMPARISON WITH BASELINES AND ABLATION STUDIES

Our method, PGNA-PL, along with various baselines, was evaluated across three datasets, as detailed in the upper sections of Tables 1-3. For each dataset, performance comparisons are provided based on two different candidate label generation approaches, namely Semantic Distribution and Russell Distribution. We compare leading algorithms in both PLL methodologies, including the IBS algorithms CR (Wu et al., 2022), CAVL (Zhang et al., 2021), PRODEN (Lv et al., 2020), LW (Wen et al., 2021), PiCO (Wang et al., 2022), and PaPi (Xia et al., 2023), and a special algorithm DNPL (Seo & Huh, 2021) . Descriptions of the baselines are available in Appendix A.6.

According to these tables, our method consistently achieves state-of-the-art results on all datasets. Notably, recent research has focused on IBS methods. For instance, PiCO updates candidate labels using a moving average of prototypes, while PaPi aligns the classification capabilities of updated candidate labels and prototypes based on classifier outputs. While these approaches also leverage prototypes to bolster model performance, their frequent updates to candidate labels can result in unstable training. In contrast, DNPL consistently achieves high performance across all three datasets

Table 1: Performance evaluation on the SEED dataset (%)

| | Semantic Distribution | | | Russel Distribution | | |
|---|---|---|---|---|---|---|
| **Method** | **Accuracy** | **Macro F1** | **Micro F1** | **Accuracy** | **Macro F1** | **Micro F1** |
| CR (Wu et al., 2022) | 54.69(±13.29) | 49.04(±17.07) | 55.07(±13.18) | 53.62(±12.86) | 47.34(±16.47) | 53.87(±12.91) |
| CAVL (Zhang et al., 2021) | 59.66(±19.05) | 51.5(±24.3) | 59.53(±18.92) | 56.29(±20.48) | 46.27(±26.5) | 55.67(±20.46) |
| PRODEN (Lv et al., 2020) | 78.04(±12.44) | 76.09(±14.1) | 77.63(±12.67) | 75.5(±13.46) | 73.1(±15.41) | 74.98(±13.72) |
| LW (Wen et al., 2021) | 78.36(±12.31) | 76.64(±13.77) | 78.02(±12.52) | 78.2(±12.54) | 76.39(±14.03) | 77.77(±12.77) |
| PiCO (Wang et al., 2022) | 78.0(±12.28) | 76.28(±13.67) | 77.61(±12.46) | 78.2(±12.32) | 76.65(±13.39) | 77.81(±12.48) |
| PaPi (Xia et al., 2023) | 77.18(±12.5) | 75.19(±14.01) | 76.76(±12.7) | 75.5(±13.12) | 72.93(±15.25) | 74.93(±13.4) |
| DNPL (Seo & Huh, 2021) | 78.27(±12.39) | 76.38(±14.05) | 77.87(±12.62) | 77.63(±12.95) | 75.75(±14.38) | 77.21(±13.14) |
| **PGNA-PL** | **79.69**(±11.86) | **78.11**(±13.11) | **79.31**(±12.07) | **79.86**(±12.3) | **77.76**(±14.24) | **79.33**(±12.6) |
| w/o PG | 78.3(±12.41) | 76.51(±13.76) | 77.88(±12.64) | 78.98(±11.98) | 77.27(±13.35) | 78.56(±12.23) |
| w/o NA | 79.09(±12.52) | 77.43(±13.9) | 78.71(±12.73) | 79.54(±12.6) | 77.53(±14.27) | 79.02(±12.84) |

Table 2: Performance evaluation on the SEED-IV dataset (%)

| | Semantic Distribution | | | Russel Distribution | | |
|---|---|---|---|---|---|---|
| **Method** | **Accuracy** | **Macro F1** | **Micro F1** | **Accuracy** | **Macro F1** | **Micro F1** |
| CR (Wu et al., 2022) | 47.63(±13.41) | 40.12(±16.16) | 46.31(±13.11) | 46.94(±12.34) | 37.92(±15.23) | 45.47(±12.03) |
| CAVL (Zhang et al., 2021) | 44.14(±14.7) | 30.6(±17.65) | 41.68(±14.41) | 43.29(±15.75) | 31.96(±20.14) | 42.0(±16.14) |
| PRODEN (Lv et al., 2020) | 63.81(±14.0) | 61.26(±15.19) | 62.98(±13.89) | 60.92(±14.7) | 57.49(±16.99) | 60.11(±14.72) |
| LW (Wen et al., 2021) | 64.11(±15.21) | 61.86(±16.69) | 63.45(±15.17) | 64.13(±15.33) | 61.38(±17.54) | 63.55(±15.25) |
| PICO (Wang et al., 2022) | 66.15(±13.4) | 64.63(±13.83) | 65.67(±13.07) | 65.33(±13.19) | 63.9(±13.34) | 64.97(±12.84) |
| PaPi (Xia et al., 2023) | 67.65(±12.5) | 65.9(±12.81) | 67.04(±12.24) | 65.96(±11.97) | 63.73(±12.46) | 65.18(±11.63) |
| DNPL (Seo & Huh, 2021) | 67.24(±12.7) | 66.01(±13.08) | 66.9(±12.54) | 67.1(±11.98) | 65.86(±12.45) | 66.88(±11.77) |
| **PGNA-PL** | **68.31**(±12.55) | **67.02**(±12.69) | **67.86**(±12.11) | 67.25(±11.73) | 65.15(±12.39) | 66.6(±11.43) |
| w/o PG | 67.74(±12.69) | 66.59(±12.89) | 67.37(±12.5) | **68.01**(±12.32) | **66.74**(±12.65) | **67.76**(±11.82) |
| w/o NA | 67.1(±12.8) | 65.71(±13.15) | 66.66(±12.42) | 65.85(±12.46) | 63.53(±13.39) | 65.1(±12.26) |

with a straightforward classification loss. PGNA-PL further improves the performance through its novel prototype-guided and noise augmentation techniques.

In the sections below the horizontal lines of the three tables, we ablated two key optimization elements: prototype guidance (w/o PG) and noise augmentation methods (w/o NA). We observed a decrease in model performance across the majority of datasets and experimental configurations when either of these modules was removed. When both modules were ablated simultaneously, the model reverted to its baseline form, DNPL, thereby validating the effectiveness of the proposed enhancements. However, an outlier was observed in Russell Distribution on the SEED-IV dataset, as detailed in Table 2. Here, our approach did not significantly surpass DNPL, and performance actually improved upon the removal of the PG module. The likely reason is that the Russell Distribution fails to accurately reflect the true semantic and physiological relationships between specific emotions in the SEED-IV dataset, leading to biased calculations of label similarity. During the self-distillation process, these biases misdirect the learning trajectory of the classifier, resulting in decreased performance. In contrast, Semantic Distribution methods such as GloVe vectors capture the semantic relationships between emotions more accurately, thereby enhancing classification results.

Additionally, to validate the advantages of our method under fully supervised conditions, relevant experiments are described in Appendix A.7. To verify our advantages over other encoders commonly used for EEG-based emotion recognition, we have provided comparative experiments with the MLP encoder, as shown in the Appendix A.8. To demonstrate the effectiveness of the proposed noise augmentation method, a comparison of the performance of PGNA-PL with related noise augmentation methods and the original mixup approach is provided in the Appendix A.9.

### 3.2.2 CONFUSION MATRICES STUDIES

To further examine the differences in how the PGNA-PL model recognizes various emotions, we analyzed the confusion matrices, as depicted in Figure 2. Subfigures (a), (b), and (c) represent the performance across three datasets under Semantic Distribution, while subfigures (d), (e), and (f) reflect performance under Russell Distribution. Across both distributions, the PGNA-PL model demonstrated exceptional ability to recognize fear, notably achieving over 70% accuracy on the SEED-V dataset, underscoring the high potential applicability in assisting in the recognition of fear-related disorders.

Table 3: Performance evaluation on the SEED-V dataset (%)

| Method | Semantic Distribution | | | Russel Distribution | | |
|---|---|---|---|---|---|---|
| | Accuracy | Macro F1 | Micro F1 | Accuracy | Macro F1 | Micro F1 |
| CR (Wu et al., 2022) | 39.21(±12.42) | 28.95(±12.84) | 37.51(±10.95) | 38.29(±12.39) | 28.07(±12.72) | 36.75(±10.59) |
| CAVL (Zhang et al., 2021) | 44.17(±18.31) | 34.66(±20.99) | 42.47(±17.91) | 38.56(±17.81) | 31.41(±19.72) | 40.26(±16.8) |
| PRODEN (Lv et al., 2020) | 54.61(±15.98) | 50.21(±16.25) | 53.67(±15.31) | 51.41(±16.22) | 48.13(±16.06) | 51.82(±14.92) |
| LW (Wen et al., 2021) | 57.59(±16.39) | 54.64(±16.7) | 57.57(±15.26) | 55.77(±17.48) | 53.84(±17.31) | 56.47(±16.03) |
| PiCO (Wang et al., 2022) | 58.92(±15.73) | 57.18(±15.29) | 59.37(±14.31) | 55.79(±16.51) | 54.35(±15.68) | 56.81(±14.65) |
| PaPi (Xia et al., 2023) | 59.59(±14.92) | 56.86(±14.81) | 59.49(±13.54) | 54.13(±15.44) | 51.87(±15.03) | 55.01(±13.87) |
| DNPL (Seo & Huh, 2021) | 59.16(±16.19) | 57.34(±15.96) | 59.64(±14.96) | 58.61(±16.22) | **57.29**(±16.05) | 59.54(±15.0) |
| **PGNA-PL** | **60.11**(±16.42) | **58.02**(±16.32) | **60.4**(±15.24) | **58.97**(±16.15) | 57.23(±16.26) | **59.78**(±14.9) |
| w/o PG | 58.11(±16.28) | 56.17(±16.15) | 58.69(±15.15) | 58.26(±16.29) | 56.95(±15.99) | 59.26(±14.88) |
| w/o NA | 59.18(±16.31) | 56.6(±16.47) | 59.14(±15.49) | 57.49(±16.85) | 55.41(±16.91) | 58.12(±15.56) |

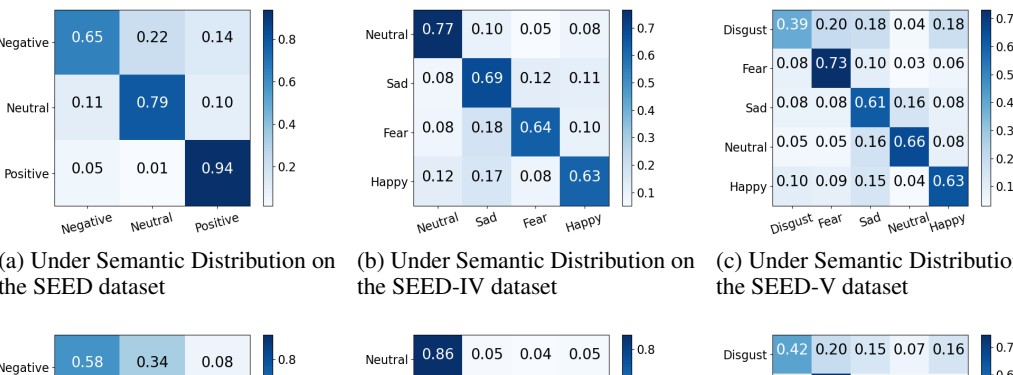

(a) Under Semantic Distribution on the SEED dataset

(b) Under Semantic Distribution on the SEED-IV dataset

(c) Under Semantic Distribution on the SEED-V dataset

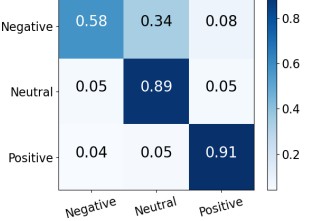
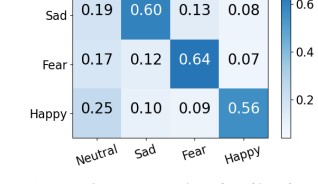
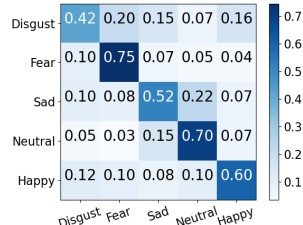

(d) Under Russel Distribution on the SEED dataset

(e) Under Russel Distribution on the SEED-IV dataset

(f) Under Russel Distribution on the SEED-V dataset

Figure 2: Confusion matrix for PGNA-PL across SEED, SEED-IV, and SEED-V datasets under Semantic Distribution and Russell Distribution.

A comparative analysis of the two distributions revealed that the recognition accuracy for sad was significantly better under Semantic Distribution than under Russell Distribution. This improvement can likely be attributed to the semantic-based candidate label generation method, which more accurately captures the nuances of sad, in contrast to Russell's distribution, where the proximity of sad and happy is zero—unrealistically extreme. Therefore, we propose that Semantic Distribution offers a more valid approach for evaluating scenarios with partial labels.

### 3.2.3 VISUALIZATION OF PROTOTYPE-GUIDED CAPABILITIES

Given that SEED-V is a dataset with five categories, including three distinct negative emotions, it is well-suited for the visual evaluation of prototype-guided effects. To visually assess the guiding capability of prototypes, we represent the similarities based on the cosine distances between emotion label features generated from two distributions on the SEED-V dataset in subfigures (a) and (b) of Figure 3, respectively. Furthermore, we employ Principal Component Analysis (PCA) (Maćkiewicz & Ratajczak, 1993) to depict the similarity between different emotion category prototypes in two scenarios, as illustrated in subfigures (c) and (d). The numbers on the line segments represent the distances between emotion categories, with greater distances indicating lower similarity.

Initially, subfigures (c) and (d) reveal common patterns between the two distributions: the three negative emotions—fear, sad, and disgust—are closely clustered, with fear and sad being the furthest

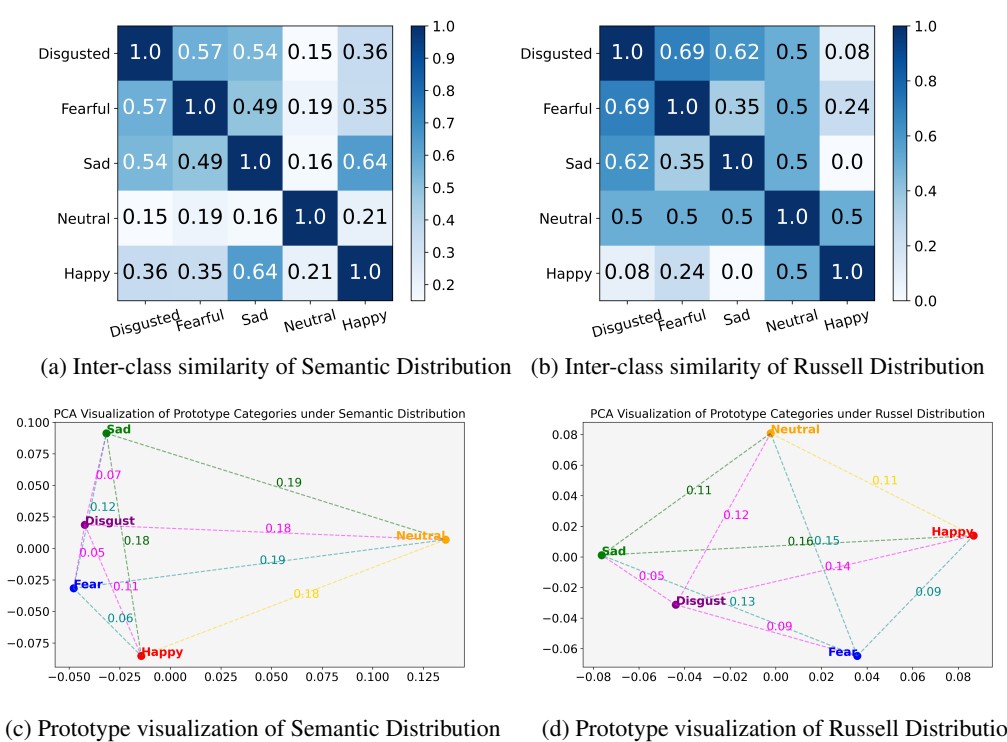

(a) Inter-class similarity of Semantic Distribution    (b) Inter-class similarity of Russell Distribution

(c) Prototype visualization of Semantic Distribution    (d) Prototype visualization of Russell Distribution

Figure 3: Inter-class similarity and generated prototypes of two distributions on the SEED-V dataset.

apart, while disgust is approximately equidistant from the other two. However, distinct differences are evident in the Semantic Distribution, as shown in subfigure (c), where compared to neutral, the other four emotion-laden categories are closer together, reflecting stronger relationships between categories with strong emotional expressions, which is characteristic of the Semantic Distribution's inter-emotional distances. Jiang et al. (2024) noted that EEG responses to positive and negative emotions are more pronounced in the frontal and parietal lobes, while responses to neutral emotions are less intense. This suggests that physiologically, the EEG responses to emotional states are more closely aligned than those to neutral states, consistent with the similarity between emotions in the Semantic Distribution. In contrast, in the Russell Distribution depicted in subfigure (d), neutral is closer from other emotions, but happy and sad are significantly farther apart, aligning with the setup of emotional similarities in subfigure (d). Overall, the prototypes effectively approximate the hypothesized distances between emotion categories under different distributions, thus suitably guiding classifiers in learning inter-class relationships and enhancing interpretability.

## 4    CONCLUSION

To address the challenges of PLL in EEG-based emotion recognition, we introduced a semantic-based method for generating candidate labels. Leveraging the GloVe vectors, this approach enhances the clarification of semantic relationships between different emotions, closely mirroring real-world scenarios of partial labeling. We also developed the PGNA-PL model, specifically designed for EEG emotion recognition. This model incorporates a self-distillation strategy, using prototypes to guide the classifier's disambiguation efforts by tapping into the inter-class distance recognition capabilities. Moreover, we crafted a tailored noise augmentation technique to address the low signal-to-noise ratio inherent in EEG data. Experiments on the SEED, SEED-IV, and SEED-V datasets, employing two distinct candidate label generation methods, confirmed that our approach achieves SOTA results. The confusion matrix suggests our method's potential superiority in helping to identify fear-related disorders.

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

# A APPENDIX

## A.1 RELATED WORK

EEG-based emotion recognition is a promising field. With advancements in basic problems such as feature extraction (Duan et al., 2013) and cross-subject emotion transfer (Wang et al., 2024a), partial label scenarios in EEG emotion recognition have gained attention. Zhang & Etemad (2023) compared classical PLL methods on the SEED-V dataset. However, how to simulate reasonable candidate labels to evaluate different PLL models in emotion recognition scenarios, and how to customize EEG emotion recognition models under PLL settings, remain areas worth exploring.

Constructing partial-label datasets in real-world scenarios is challenging, as it requires both a training set with uncertain labels and a test set with fully certain labels (Cour et al., 2011). Current methods (Wen et al., 2021; Wang et al., 2022) typically simulate candidate labels in the training set by introducing known erroneous labels into standard datasets (e.g., using a uniform distribution to assign candidate labels), but they neglect the correlations between labels. Zhang & Etemad (2023) addressed this for emotion labels by using the Russell's circumplex emotion model (Russell, 1980) to compute relationships among emotions for generating candidate labels, yet their method did not fully exploit semantic relationships between emotion categories.

PLL models, particularly identification-based strategies (IBS) (Lv et al., 2023) that optimize candidate labels during training, have been well explored in recent years (Lv et al., 2020; Wen et al., 2021; Zhang et al., 2021; Wu et al., 2022; Wang et al., 2022; Xia et al., 2023). such as PiCO (Wang et al., 2022) and PaPi (Xia et al., 2023), enhance model predictions by iteratively updating candidate labels with the help of the prediction results of prototypes. DNPL (Seo & Huh, 2021), distinct from traditional IBS, dynamically adjusts both candidate and non-candidate label probabilities in response to model predictions, enhancing learning accuracy and robustness without optimize candidate labels. This approach helps DNPL avoid overfitting to incorrect labels, making it a unique method within the PLL framework. However, further exploration of model enhancement techniques for EEG-based emotion recognition, including self-distillation (Zhang et al., 2019) and noise augmentation (Wang et al., 2024b), remains an evolving area of research.

Self-distillation leverages a single-step approach that focuses its training efforts directly on the student model (Zhang et al., 2019), unlike traditional knowledge distillation (Hinton, 2015) which requires a pre-trained teacher model to guide the student. For PLL in EEG-based emotion recognition, Li et al. (2024a) replicate an identical model as the teacher, which, while effective, greatly increases parameters. In contrast, we use a single prototype as the teacher, avoiding this overhead. Moreover, although noise augmentation techniques have been explored to some extent in EEG-based emotion recognition, such as the time steps shuffling method proposed by Wang et al. (2024b) to introduce perturbations within the features of a time window, there is still no well-established approach for applying noise augmentation to the fusion of individual EEG features.

## A.2 Introduction to Relevant Concepts

### A.2.1 Description of GloVe vectors (840B Tokens, 2.2M Vocabulary Version)

Global Vectors for Word Representation (GloVe) is a widely utilized resource in the field of natural language processing. It encodes words into numerical vectors using an unsupervised learning algorithm that leverages global word-word co-occurrence statistics from a corpus.

The version used in this study is the largest available, encompassing data derived from 840 billion tokens. It contains a vocabulary of 2.2 million unique, case-sensitive words and provides pre-trained vectors of 300 dimensions for each word. The vectors capture not only the distributional semantics of words but also subtle semantic relationships and patterns in the data. These vectors, totaling 2.03 GB, are instrumental in various applications, including but not limited to text analysis, affective computing, and machine learning models that rely on robust word representations.

### A.2.2 Description of Russel Distribution

The real-world experimental framework referred to as the Russell Distribution in this paper, introduces a novel candidate label generation technique in partial label learning tailored for emotion recognition tasks. This method leverages Russell's circumplex model, where emotions are mapped on a circular layout according to arousal and valence. Each emotion is defined by its polar coordinates—radius and angle.

To generate candidate labels, the method calculates these polar coordinates for each emotion. It then uses them to determine a normalized similarity score between emotions based on the cosine of their angular differences and their Euclidean distances on the wheel.

Instead of using a fixed probability for a class to become a candidate label, this approach dynamically adjusts the probabilities based on the similarity scores. Emotions closer to the true label on the wheel are more likely to be chosen as candidate labels.

This method offers a more realistic way of generating candidate labels by acknowledging the natural similarities between emotions, thus minimizing the risk of confusing similar emotions and enhancing the learning effectiveness.

## A.3 MODEL ARCHITECTURE

As shown in Table 4, the encoder consists of four blocks: convolutional layer 1, convolutional layer 2, Transition, and the fully connected block, concatenated in the order described. The classifier is a fully connected network that maps the features to the number of emotion classes, thereby outputting the probability distribution for emotion recognition.

Table 4: Model architecture overview

| Basic Module | Block | Layer | Description/Hyperparameters |
|---|---|---|---|
| Encoder | Conv Layer 1 | Conv1D | In_channels=1, Out_channels=5, Kernel_size=3, Stride=1 |
| | | BatchNorm | Normalize across 5 feature channels |
| | | LeakyReLU | Activation with Negative_slope=0.3 |
| | Conv Layer 2 | Conv1D | In_channels=5, Out_channels=10, Kernel_size=3, Stride=1 |
| | | BatchNorm | Normalize across 10 feature channels |
| | | LeakyReLU | Activation with Negative_slope=0.3 |
| | Transition | Flatten | Flattens output for fully connected layers |
| | Fully Connected Block | Linear | In_features=3060, Out_features=64 |
| | | ReLU | Fully connected activation |
| | | Dropout | Dropout with p=0.5 |
| Classifier | Final Fully Connected Layer | Linear | In_features=64, Out_features=number of emotions |

## A.4 DATASETS

**SEED Dataset**. The SEED dataset introduces three distinct emotions (positive, neutral, and negative), represented by 15 film clips. A total of 15 participants (7 males and 8 females) engaged in the experiment, completing three trials for each emotion across three sessions, with the same stimuli presented in each session. The primary goal of this dataset is to investigate emotional responses, making it suitable for studies focused on binary or ternary emotion classification. The limited number of emotion categories and film clips allows for simpler yet effective emotion analysis. In this work we use all three emotion categories for evaluation.

**SEED-IV Dataset**. The SEED-IV dataset introduces four distinct emotions (happy, sad, neutral, and fear), represented by 72 film clips. This dataset includes 15 participants, each completing 24 trials (6 per emotion) across three sessions. SEED-IV allows for the exploration of a broader emotional spectrum compared to the SEED dataset, providing a richer dataset for multi-class emotion classification.

**SEED-V Dataset**. The SEED-V dataset introduces five distinct emotions (happy, neutral, sad, fear, and disgust), represented by 45 short films. A total of 16 participants (10 females and 6 males) took part in the experiment, each completing three trials for each emotion over three sessions, with entirely new stimuli presented in each session. The addition of the disgust emotion extends the emotional granularity of the dataset, making SEED-V particularly suitable for nuanced emotion recognition tasks and enabling more complex models to classify a wider range of emotions.

For all datasets (SEED, SEED-IV, SEED-V), EEG recordings were collected using a 62-channel ESI NeuroScan System. The EEG data were divided into non-overlapping 4-second segments for analysis. Differential Entropy (DE) features were extracted from five EEG frequency bands: $\delta$ (1-4 Hz), $\theta$ (4-8 Hz), $\alpha$ (8-14 Hz), $\beta$ (14-31 Hz), and $\gamma$ (31-50 Hz). A total of 310 dimensions of DE features were obtained for each segment across all channels. To prepare for model training, we normalized the vector of each DE feature using the Min-Max normalization method (Patro, 2015), applied across the entire dataset, to scale all feature values to fall between specified minimum and maximum values, which in this text are 0 and 1. As shown in the formula below, where $\max(X)$ is the maximum value and $\min(X)$ is the minimum value, and $X'$ is the normalized data.

$$X' = \frac{X - \min(X)}{\max(X) - \min(X)} \tag{14}$$

## A.5 HYPERPARAMETER SETTINGS

To ensure fair alignment with baseline methods, we fixed the common hyperparameters across different models. Table 5 lists the common hyperparameters between models and the hyperparameters specific to our model's unique modules. We determined the optimal values of $\beta_p$ for various distributions across different datasets through a grid search method. The values of $\beta_p$ were taken from the range [0.5, 4], with an interval of 0.5 between points. The corresponding hyperparameters are also listed in Table 5. All evaluations are performed over five independent runs with 5 different random seeds, with results averaged. Experiments are executed on a single NVIDIA Tesla V100 GPU using PyTorch.

Table 5: Hyperparameters description

| Hyperparameters Category | Hyperparameter | Value |
|---|---|---|
| Common | Batch size | 8 |
| | Number of epochs | 30 |
| | Learning rate | 0.01 |
| | Dimensionality ($d$) | 64 |
| | Optimizer | SGD |
| | SGD momentum | 0.9 |
| | SGD decay rate | 0.0001 |
| PGNA-PL | Prototype optimization ($\xi$) | 0.99 |
| | Self-distillation temperature ($\tau$) | 2 |
| | Balance parameter ($\alpha$) | 0.5 |
| | Controlled noise augmentation ($\sigma$) | 0.8 |
| $\beta_p$ for PGNA-PL | under Semantic Distribution on the SEED dataset | 3 |
| | under Russel Distribution on the SEED dataset | 3 |
| | under Semantic Distribution on the SEED-IV dataset | 2 |
| | under Russel Distribution on the SEED-IV dataset | 0.5 |
| | under Semantic Distribution on the SEED-V dataset | 3.5 |
| | under Russel Distribution on the SEED-V dataset | 3 |

## A.6 BASELINES METHODS

We provide a detailed overview of classical IBS methods and the DNPL approach as follows:

### A.6.1 IBS METHODS

- **PRODEN (Lv et al., 2020):** PROgressive iDENtification (PRODEN) updates the label sets based on their compatibility with model predictions, iteratively refining this relationship. By directly adjusting the labels in response to prediction accuracy, the method fosters model convergence on the correct labels amid noisy or partial label information.

- **LW (Wen et al., 2021):** Leveraged Weighted (LW) loss introduces a novel approach in partial label learning by incorporating a leverage parameter $\beta$, which allows for a strategic balance between losses on candidate and non-candidate labels. LW loss not only provides a method to adjust the influence of different types of labels dynamically, but it is also backed by a theoretical framework that assures risk consistency under relatively weak assumptions. The theoretical foundations of LW loss offer guidance on the optimal choice of $\beta$, enhancing the method's effectiveness and adaptability in diverse learning environments.

- **CAVL (Zhang et al., 2021):** Class Activation Value Learning (CAVL) is a novel method in partial-label learning that eschews the usual assumptions about data collection that can limit the performance of traditional models. CAVL uses a technique called Class Activation Value (CAV), which adapts the principles of Class Activation Maps (CAM) from image-based neural networks to be applicable across various input types and models. This approach allows CAVL to identify the true label from a set of candidates by selecting the

class with the highest CAV, thereby promoting accurate model training without relying on predefined data assumptions.

- **CR (Wu et al., 2022) :** The method leverages a novel training framework combining supervised learning on non-candidate labels with consistency regularization on candidate labels, addressing the challenge of PLL. By integrating consistency regularization, the method aligns predictions across multiple augmented versions of the same instance to infer a reliable label distribution. This alignment ensures the model learns robust representations while preserving consistency in predictions under different input perturbations. The inferred label distribution is adaptively optimized using a closed-form solution, enabling an efficient and effective approach to guide the model towards the true label distributions. This framework enhances the learning process by utilizing non-candidate labels to refine model predictions while maintaining stability and adaptability through regularization, distinguishing it from traditional methods reliant on self-training or contrastive learning.

- **PiCO (Wang et al., 2022):** Partial label learning with COntrastive label disambiguation (PiCO) leverages class prototypes to effectively address two fundamental challenges in partial label learning (PLL): representation learning and label disambiguation. By using the classification outcomes of prototypes, it refines the candidate labels through a prototype-based disambiguation process, aligning the label prediction with the most likely true class. To enhance representation learning, the method incorporates a contrastive learning module that combines MoCo (He et al., 2020) and SupCon (Khosla et al., 2020) approaches. This module promotes closely aligned representations for examples within the same class while maintaining separation between different classes. The integration of these components into a unified framework not only facilitates better disambiguation of ambiguous labels but also boosts the encoder's ability to generate high-quality representations. The method can be rigorously explained from an expectation-maximization (EM) perspective, underscoring the mutual reinforcement between representation learning and label disambiguation.

- **PaPi (Xia et al., 2023):** Partial-label learning with a guided Prototypical classifier (PaPi) directly employs classifier results for disambiguation and aligns prototype classifications with candidate label distributions using KL divergence, showcasing several notable characteristics. By leveraging a shared feature encoder between a linear classifier and prototypical representations, the approach effectively encourages the feature space to reflect the intrinsic similarities between categories. This alignment enhances the model's ability to distinguish between candidate labels by explicitly guiding prototype optimization through the predictive results of the classifier. Unlike methods relying on contrastive learning, which can introduce noise and require significant computational resources, this approach avoids such dependencies, providing a more streamlined and computationally efficient framework for partial-label learning. By focusing on directly aligning prototype classifications with candidate label distributions, the method not only simplifies the disambiguation process but also strengthens representation learning in ambiguous label scenarios.

### A.6.2 THE DNPL METHOD

- **DNPL (Seo & Huh, 2021):** Deep Naive Partial label Learning (DNPL) revolutionizes traditional PLL approaches by dynamically integrating both candidate and non-candidate label probabilities into the model training process. Unlike standard PLL methods that rely on static label disambiguation, DNPL leverages a sophisticated algorithm to dynamically adjust the influence of each label based on its correlation with model predictions. This method effectively minimizes overfitting to incorrect candidate labels, enhances the robustness of learning, and allows the model to better discern the true label within noisy or ambiguous candidate sets.

### A.7 EFFECTIVENESS UNDER FULL SUPERVISION

This paper introduces two modules, prototype-guided (PG) and noise augmentation (NA), designed for learning inter-class relationships and enhancing the noise resistance of EEG classifiers, respectively. These issues are extended to scenarios under full supervision, where authentic labels are used as classification targets. To demonstrate the advantages of our proposed PGNA-PL method under full supervision, we provide a comparative analysis in Table 6 of the performances between PGNA-

PL under full supervision and the results obtained using only the backbone and classifier described in this paper. Specifically, under full supervision, both PGNA-PL and the comparative method employ cross-entropy loss for the classification loss $L_{CLS}$. In terms of hyperparameters, consistent hyperparameters were applied across the experiments under the condition of Semantic Distribution.

Table 6: Comparison of performance metrics under full supervision on the SEED, SEED-IV, and SEED-V datasets

| Method | SEED | | | SEED-IV | | | SEED-V | | |
| --- | --- | --- | --- | --- | --- | --- | --- | --- | --- |
| | Accuracy | Macro F1 | Micro F1 | Accuracy | Macro F1 | Micro F1 | Accuracy | Macro F1 | Micro F1 |
| Backbone (full supervision) | 78.05 (±12.68) | 76.22 (±14.05) | 77.63 (±12.91) | 67.67 (±11.8) | 66.37 (±12.08) | 67.44 (±11.47) | 59.29 (±15.49) | 59.35 (±14.57) | 60.06 (±14.39) |
| **PGNA-PL (full supervision)** | **80.11 (±12.02)** | **78.47 (±13.4)** | **79.7 (±12.25)** | **68.8 (±12.08)** | **67.49 (±12.29)** | **68.36 (±11.75)** | **61.02 (±16.14)** | **59.4 (±16.04)** | **61.62 (±14.77)** |

The data from Table 6 indicates that under full supervision conditions, PGNA-PL outperforms the backbone across all three datasets. This suggests that the two proposed modules are generalizable to the broader problem of EEG emotion recognition.

### A.8 COMPARISON OF ENCODER REPLACEMENT WITH MLP

To further validate our method, we compared the effect of replacing the encoder with a commonly used alternative multilayer perceptron (MLP) (Li et al., 2024a; Zhou et al., 2023; Li et al., 2018; Chen et al., 2021; Li et al., 2021) against our method and the baselines. This encoder follows a structure similar to that used by Li et al. (2024a), with input, hidden, and output dimensions of 310, 256, and 64 respectively, and employs the ReLU activation function. Our comparisons were conducted on the SEED, SEED-IV, and SEED-V datasets, keeping most hyperparameters consistent while performing a re-grid search for $\beta_p$ within the range of [0.5-4], as detailed in Table 7. The evaluation metrics remained consistent with those used in our study. Results are detailed in Tables 8-10.

Table 7: Optimal $\beta_p$ across Different Datasets and Distributions

| **Dataset** | **Semantic Distribution** | **Russel Distribution** |
| --- | --- | --- |
| SEED | 4 | 1 |
| SEED-IV | 1 | 1 |
| SEED-V | 2.5 | 4 |

Table 8: Performance evaluation using an MLP encoder on the SEED dataset (%)

| | **Semantic Distribution** | | | **Russel Distribution** | | |
| --- | --- | --- | --- | --- | --- | --- |
| **Method** | **Accuracy** | **Macro F1** | **Micro F1** | **Accuracy** | **Macro F1** | **Micro F1** |
| CR (Wu et al., 2022) | 52.57(±13.51) | 45.45(±17.13) | 53.03(±13.42) | 52.83(±14.62) | 45.26(±18.71) | 53.04(±14.68) |
| CAVL (Zhang et al., 2021) | 77.78(±12.84) | 75.61(±14.72) | 77.38(±13.07) | 77.6(±12.89) | 75.61(±14.64) | 77.23(±13.08) |
| PRODEN (Lv et al., 2020) | 78.26(±12.17) | 75.94(±14.21) | 77.82(±12.42) | 75.97(±13.11) | 73.41(±15.21) | 75.46(±13.32) |
| LW (Wen et al., 2021) | 77.84(±12.92) | 75.54(±14.95) | 77.4(±13.17) | 77.89(±12.62) | 75.58(±14.61) | 77.43(±12.84) |
| PiCO (Wang et al., 2022) | **78.6(±12.02)** | **76.41(±14.19)** | **78.22(±12.26)** | 77.97(±12.23) | 75.83(±14.14) | 77.54(±12.46) |
| PaPi (Xia et al., 2023) | 75.85(±12.64) | 73.08(±15.09) | 75.38(±12.85) | 72.18(±13.94) | 68.48(±16.86) | 71.54(±14.16) |
| DNPL (Seo & Huh, 2021) | 77.21(±13.0) | 74.87(±15.2) | 76.77(±13.25) | 77.42(±13.09) | 75.27(±14.93) | 77.0(±13.28) |
| **PGNA-PL** | 77.92(±13.08) | 75.85(±14.9) | 77.54(±13.27) | **78.35(±12.93)** | **75.96(±15.19)** | **77.85(±13.19)** |

Table 9: Performance evaluation using an MLP encoder on the SEED-IV dataset (%)

| Method | Semantic Distribution | | | Russel Distribution | | |
|---|---|---|---|---|---|---|
| | Accuracy | Macro F1 | Micro F1 | Accuracy | Macro F1 | Micro F1 |
| CR (Wu et al., 2022) | 45.34(±12.34) | 36.83(±14.82) | 44.18(±11.99) | 46.92(±12.72) | 37.8(±15.78) | 45.57(±12.78) |
| CAVL (Zhang et al., 2021) | 59.19(±16.92) | 51.76(±21.26) | 57.46(±17.1) | 57.07(±17.84) | 50.7(±22.23) | 56.02(±17.93) |
| PRODEN (Lv et al., 2020) | 69.48(±12.3) | 67.76(±12.79) | 68.84(±12.01) | 68.23(±12.35) | 66.77(±12.59) | 67.75(±11.91) |
| LW (Wen et al., 2021) | 69.31(±12.3) | 67.92(±12.65) | 68.92(±11.92) | 69.31(±12.3) | 68.26(±12.15) | 69.04(±11.69) |
| PiCO (Wang et al., 2022) | 69.23(±11.91) | 67.65(±12.32) | 68.79(±11.5) | 68.26(±12.4) | 67.03(±12.61) | 67.91(±11.88) |
| PaPi (Xia et al., 2023) | 67.89(±11.65) | 66.08(±12.2) | 67.39(±11.37) | 65.28(±12.67) | 63.64(±12.91) | 64.8(±12.12) |
| DNPL (Seo & Huh, 2021) | 66.99(±12.67) | 65.65(±12.97) | 66.54(±12.27) | 67.11(±12.94) | 65.77(±13.33) | 66.66(±12.53) |
| **PGNA-PL** | **69.63(±11.54)** | **68.54(±11.67)** | **69.31(±11.17)** | **69.61(±11.3)** | **68.5(±11.25)** | **69.26(±10.73)** |

Table 10: Performance evaluation using an MLP encoder on the SEED-V dataset (%)

| Method | Semantic Distribution | | | Russel Distribution | | |
|---|---|---|---|---|---|---|
| | Accuracy | Macro F1 | Micro F1 | Accuracy | Macro F1 | Micro F1 |
| CR (Wu et al., 2022) | 37.67(±12.26) | 26.56(±12.79) | 35.85(±11.12) | 36.87(±12.22) | 26.23(±12.55) | 35.42(±10.96) |
| CAVL (Zhang et al., 2021) | 57.28(±16.45) | 53.65(±17.63) | 57.29(±15.87) | 52.59(±16.73) | 50.11(±17.2) | 54.39(±14.86) |
| PRODEN (Lv et al., 2020) | 61.09(±15.87) | 59.43(±15.83) | 61.84(±14.49) | 57.54(±16.61) | 55.64(±16.17) | 58.94(±14.79) |
| LW (Wen et al., 2021) | 61.9(±16.24) | 60.42(±16.06) | 62.86(±14.82) | 60.06(±17.44) | 58.45(±17.24) | 61.28(±15.88) |
| PiCO (Wang et al., 2022) | 61.04(±16.93) | 59.5(±16.68) | 61.88(±15.48) | 58.65(±16.95) | 57.58(±16.38) | 60.1(±15.29) |
| PaPi (Xia et al., 2023) | 58.78(±15.96) | 56.68(±15.68) | 59.4(±14.45) | 54.72(±16.72) | 52.63(±16.11) | 55.97(±15.08) |
| DNPL (Seo & Huh, 2021) | 60.05(±16.66) | 58.78(±16.47) | 61.35(±15.07) | 60.13(±16.6) | 58.81(±16.23) | 61.36(±15.1) |
| **PGNA-PL** | **62.38(±16.43)** | **60.82(±16.32)** | **63.22(±14.85)** | **60.28(±16.62)** | **58.72(±16.5)** | **61.4(±15.24)** |

As shown in Tables 8-10, our proposed PGNA-PL method achieves state-of-the-art (SOTA) results across most datasets and various distributions, with only suboptimal performance on the SEED dataset under the Semantic Distribution. Compared to the CNN encoder used in the main text of this paper, the MLP encoder performs poorly on the SEED dataset but shows better results on the other two datasets. Despite the varied performance of different encoders on different datasets, our method consistently achieves SOTA in most experiments, demonstrating good generalizability across different encoders.

## A.9 COMPARISON WITH OTHER NOISE AUGMENTATION METHODS

We further compare the effects of our proposed noise augmentation method with other noise augmentation methods. The experimental results on the three datasets are shown in Tables 11-13. Here, "w/o NA" refers to the scenario without noise augmentation. Since the PGNA-PL constraint retains at least 80% of the original features, similarly, "Add Gauss noise" involves adding Gaussian noise with a standard deviation of 20%, and "Mask Features" randomly sets 20% of the features to zero. The term "Mixup" refers to the mixup method (Zhang et al., 2018), a data augmentation technique that, unlike our method, also interpolates partial labels. Across different datasets, when interpolating various samples, it utilizes the same hyperparameters as our method, such as $\beta_p$ and $\sigma$ in Equation 9-10, to ensure a fair comparison. Moreover, partial labels are interpolated in a manner similar to that described in Equation 11.

Table 11: Comparing different noise augmentation methods on the SEED dataset

| Method | Semantic Distribution | | | Russel Distribution | | |
|---|---|---|---|---|---|---|
| | Accuracy | Macro F1 | Micro F1 | Accuracy | Macro F1 | Micro F1 |
| w/o NA | 79.09(±12.52) | 77.43(±13.9) | 78.71(±12.73) | 79.54(±12.6) | 77.53(±14.27) | 79.02(±12.84) |
| Add Gauss Noise (20%) | 79.52(±11.93) | 77.66(±13.6) | 79.14(±12.18) | **80.11(±12.01)** | **78.1(±13.83)** | **79.63(±12.29)** |
| Mask 20% Features | 77.41(±12.27) | 75.4(±13.95) | 77.01(±12.49) | 78.4(±11.97) | 76.21(±13.77) | 77.9(±12.24) |
| Mixup | 79.36(±12.01) | 77.69(±13.44) | 78.98(±12.25) | 79.3(±12.45) | 77.08(±14.48) | 78.75(±12.75) |
| PGNA-PL | **79.69(±11.86)** | **78.11(±13.11)** | **79.31(±12.07)** | 79.86(±12.3) | 77.76(±14.24) | 79.33(±12.6) |

Table 12: Comparing different noise augmentation methods on the SEED-IV dataset

| Method | Semantic Distribution | | | Russel Distribution | | |
|---|---|---|---|---|---|---|
| | Accuracy | Macro F1 | Micro F1 | Accuracy | Macro F1 | Micro F1 |
| w/o NA | 67.1($\pm$12.8) | 65.71($\pm$13.15) | 66.66($\pm$12.42) | 65.85($\pm$12.46) | 63.53($\pm$13.39) | 65.1($\pm$12.26) |
| Gauss Noise (20%) | 64.22($\pm$14.68) | 62.09($\pm$15.71) | 63.52($\pm$14.43) | 63.68($\pm$15.13) | 60.73($\pm$16.8) | 62.91($\pm$14.92) |
| Mask 20% Features | 64.66($\pm$12.26) | 63.27($\pm$12.45) | 64.44($\pm$11.77) | 64.4($\pm$12.2) | 62.36($\pm$12.96) | 64.02($\pm$11.9) |
| Mixup | 68.29($\pm$11.98) | 66.95($\pm$12.19) | 67.82($\pm$11.68) | 66.92($\pm$11.89) | 64.77($\pm$12.77) | 66.32($\pm$11.61) |
| PGNA-PL | **68.31($\pm$12.55)** | **67.02($\pm$12.69)** | **67.86($\pm$12.11)** | **67.25($\pm$11.73)** | **65.15($\pm$12.39)** | **66.6($\pm$11.43)** |

Table 13: Comparing different noise augmentation methods on the SEED-V dataset

| Method | Semantic Distribution | | | Russel Distribution | | |
|---|---|---|---|---|---|---|
| | Accuracy | Macro F1 | Micro F1 | Accuracy | Macro F1 | Micro F1 |
| w/o NA | 59.18($\pm$16.31) | 56.6($\pm$16.47) | 59.14($\pm$15.49) | 57.49($\pm$16.85) | 55.41($\pm$16.91) | 58.12($\pm$15.56) |
| Gauss Noise (20%) | 57.36($\pm$17.94) | 55.39($\pm$17.24) | 57.85($\pm$16.34) | 56.21($\pm$17.27) | 54.34($\pm$16.96) | 56.99($\pm$16.0) |
| Mask 20% Features | 56.41($\pm$17.94) | 54.24($\pm$17.15) | 56.98($\pm$16.44) | 54.69($\pm$17.25) | 52.94($\pm$16.63) | 55.5($\pm$15.87) |
| Mixup | 59.58($\pm$17.09) | 57.39($\pm$16.93) | 59.93($\pm$15.81) | **59.46($\pm$16.57)** | **57.92($\pm$16.78)** | **60.38($\pm$15.33)** |
| PGNA-PL | **60.11($\pm$16.42)** | **58.02($\pm$16.32)** | **60.4($\pm$15.24)** | 58.97($\pm$16.15) | 57.23($\pm$16.26) | 59.78($\pm$14.9) |

As can be seen from the table, in the vast majority of cases, our PGNA-PL method performs optimally. The effects of Gaussian noise are lower than those without noise addition, presumably because they deviate from the noise distribution of EEG signals. The Mask Features method results in feature loss, thus also leading to decreased performance. In contrast, our controllable noise injection method provides a noise distribution similar to that of EEG signals, thus improving performance under both Semantic and Russell distributions. The Mixup method achieved results close to ours because it also applied controllable noise techniques proposed by us. However, in most cases, PGNA-PL performs better, presumably because the interpolation of partial labels by the Mixup method increases their confusion.

