# OpenReview forum: "EEG-Based Emotion Recognition via Prototype-Guided Disambiguation and Noise Augmentation in Partial-Label Learning"
_ICLR.cc/2025/Conference — Submitted to ICLR 2025_

### Official Review · Reviewer_exz1 · 2024-10-18

**Soundness:** 2
**Presentation:** 3
**Contribution:** 2
**Rating:** 5
**Confidence:** 4

**Summary:**

The paper introduces a semantic-based candidate label generation approach for addressing the challenge of Partial Label learning (PLL) in EEG-based emotion recognition by incorporating the semantic relationship of emotions. Their approach incorporated a prototype-guided module with a self-distillation strategy to improve the generalization capability.
Experiments were conducted on three benchmark affective computing datasets.

**Strengths:**

•	The overall application is interesting. The Emotion recognition task using EEG signals is challenging in the affective computing field.
•	Partial labeling learning (PLL) is an emerging framework in weakly supervised machine learning with interesting application in the affective computing. PLL can potentially model the case where the affective label is uncertain i.e. PLL can model the case where each training example corresponds to a candidate affective label set and only one label concealed in the set is the ground-truth label.

**Weaknesses:**

•	The novelty contribution of the paper is limited. The overall method seems quite specific to the task and the benchmark datasets. No theoretical and empirical evidence was provided on how the approach can be generalized on different affective computing tasks (e.g., multi-task scenario) and different naturalistic datasets
•	The SNR augmentation techniques proposed by the authors are not novel and seem completely based on the paper Zhang et al., 2018. This point cannot be a claim for contribution
•	The paper is not correctly placed in the context of related literature. It is not clear the difference concerning other partial label learning approaches and the difference concerning self-distillation methods
•	Generating Emotionally Semantic Candidate Labels: The GloVe approach was proposed by Pennington et al. 2014. The authors fully replicate the GloVe approach to obtain semantic embedding for various emotions. Moreover, emotions can be very complex (different hierarchy, scale, etc..). The authors should cover different approaches for modeling the emotion (i.e., dimensional and continuous affect modeling). This multi-dimensional nature of the emotions is not taken into account in Eq1-3
•	Prototype-Guided Noise-Augmented Partial Label Learning Model: Also here, the module seems composed of a standard decoder and classifier approach widely used in the literature
•	Architecture details: It is not clear why the authors do not explore other models architecture
•	Experimental setup: Although the authors provide details on the hyperparameters in Appendix Table 6, it is unclear how the hyperparameter tuning was performed, i.e., hyperparameter range and optimized metric.
•	Experimental setup: It is unclear why the authors do not implement a Leave-one-subject-out procedure. In the emotion recognition task, it is highly relevant that the model can generalize across unseen subjects instead of unseen trials (perhaps to the same training subject)
•	Experimental setup: experiments are limited. Multiple task case is not considered. Several emotions can occur simultaneously. Moreover, other naturalistic datasets should be considered for evaluating the proposed approach
•	Results: Tables 1,2 and 3. Standard deviation accuracy is very high. Macro-f1 and micro-f1 are reported without any statistical test. Overall, it is impossible to justify the significative gain of the proposed approach concerning other state-of-the-art methodologies.
•	Visualization of prototype-guided capabilities: this analysis tries to visualize the latent relationship among different emotion labels. However, it is unclear how this relationship is connected with the predictive performance of the proposed model.

Minor points

•	Prototype-Guided Noise-Augmented Partial Label Learning Model: authors should provide a rationale behind the feature extraction stage. It is not clear why only the differential entropy features of EEG are used
•	Notation of different equations is sometimes omitted or ill-defined (e.g. v_{j} and v_{i} in equation 1)
•	The overall organization of the paper is poor. For instance, section 3.3 provides a feature preprocessing step to increase the SNR. It should be placed before the methodological section. Moreover, there are several sentences that disrupt the logical flow of the paper

**Questions:**

•	The paper lacks a clear differentiation from existing partial label learning and self-distillation methods. Can you clarify how your approach significantly differs from these established methods?
•	Considering the complexity and multi-dimensionality of emotions, why was the GloVe method chosen exclusively for semantic embedding? Are there alternative approaches that might capture the multi-dimensional nature of emotions more effectively?
•	Could you elaborate on the process for hyperparameter tuning, including the range of hyperparameters tested and the criteria for optimization?
•	Why was the leave-one-subject-out procedure not implemented, given its relevance in demonstrating generalizability across unseen subjects in emotion recognition tasks?
•	The experiments appear limited to specific tasks and datasets. How might the model perform across multiple simultaneous emotional tasks and other naturalistic datasets?
•	Is the performance of the proposed approach significantly better than that of other competitors?

---

> ### Author Response · Authors · 2024-12-02
> **Response to Reviewer exz1**
>
> **W1:** The approach is limited in novelty, task-specific, and lacks evidence of generalizability to other affective computing tasks, like multi-task scenarios or naturalistic datasets.
>
> **Reply:** PLL for EEG-based emotion recognition is a relatively new technique with limited research available. In terms of innovation, we are the first to simulate a candidate label generation method based on GloVe Vectors for PLL learning in the EEG emotion recognition domain, offering a new noisy label generation approach for the affective computing field. Additionally, we propose a custom model for EEG emotion recognition, which uses prototypes to help the classifier learn the distance relationships between emotion labels, and incorporates a mixup-based noise augmentation module to address the low signal-to-noise ratio (SNR) issue inherent in EEG data. To further demonstrate the generalization of our approach, we conducted a comparison of the proposed modules under a fully supervised setup in Appendix A.7 (lines 912-936), showing that the model still provides benefits even in a fully supervised scenario. This experiment suggests that our method may be effective in a wider range of tasks. Regarding your comments on multi-task scenarios and natural datasets, current PLL experiments, such as PiCO and other classic methods, have been conducted on single-task datasets. Particularly since EEG is easily disturbed by noise, to our knowledge, almost all related EEG datasets are collected under strict laboratory conditions. We will continue to focus on the development of relevant datasets and attempt to extend our model and methods to these datasets in future work.
>
> **W2:** The SNR augmentation technique appears to be based on Zhang et al., 2018 and lacks novelty.
>
> **Reply:** We are the first to adapt the Mixup method for denoising tasks and have made two key improvements for PLL learning:
> 1. We impose a constraint on the Mixup ratio using Equation 10, ensuring that the original samples retain more of their features. Through hyperparameter random search, we found that the optimal range for $\lambda'$ is [0.8, 0.9]. These methods and experiments ensure the controllability of the introduced noise.
> 2. Unlike mixup, which interpolates the labels of the newly generated data, we consider that doing so under the PLL problem might further confuse the labels. Instead, we keep the original labels unchanged. This approach aligns with the noise addition constraint, rather than data augmentation. To further demonstrate this second point, we added a comparison with the mixup method under the same parameter conditions in Appendix A.9 (lines 1006-1053), where we found that our method outperforms mixup in most cases. These improvements and experiments provide sufficient evidence to support the novelty and contribution of our noise addition method.
>
> **W3:** The paper does not clearly position itself within the related literature, especially regarding partial label learning and self-distillation methods.
>
> **Reply:** We have moved the Related Work section to Appendix A.1 and highlighted the differences between our approach and other PLL learning methods in lines 710-719. Additionally, in Appendix A.6 (lines 842-910), we provide a more detailed description of various baseline partial label learning methods. Regarding self-distillation methods, we have also added an explanation in the Related Work section, as detailed in lines 721-728.
>
> **W4:** The use of GloVe for generating emotionally semantic candidate labels replicates Pennington et al., 2014. The multi-dimensional nature of emotions (e.g., hierarchical, continuous models) is not considered in the approach.
>
> **Reply:** The basic assumption of PLL is that only one label is correct among several candidate labels. Therefore, datasets related to dimensional and continuous emotion modeling, such as DEAP, do not fit the assumptions of PLL learning. When these datasets are used for classification, such as cutting continuous labels along the valence dimension to create binary or ternary classifications, these categories mainly reflect different intensities along the Valence dimension, but there is no inherent relationship between them like that between distinct emotional categories. Thus, they are not suitable for the semantic candidate label generation approach we propose. However, in terms of the model, experiments in Appendix A.7 (lines 912-936) show that even under fully supervised supervision, the two modules we proposed, PG and NA, still provide benefits. This suggests that under a fully supervised assumption, our method may generalize to more datasets, especially the noise augmentation module, which addresses the inherent low SNR issue of EEG signals—an issue that is also present in the DEAP dataset. Therefore, while PLL learning methods are currently challenging to apply to dimensional and continuous emotion-related datasets, the methods we propose could be beneficial for such datasets.

---

> ### Author Response · Authors · 2024-12-02
> **Response to Reviewer exz1 (Cont'd)**
>
> **W5:** The Prototype-Guided Noise-Augmented Partial Label Learning model appears to rely on standard decoder and classifier architectures commonly used in the literature.
>
> **Reply:** Our method does not use a decoder. The approach you referred to appears to be similar to the denoising autoencoder method in MAE [1]. This method reconstructs features of the masked (invisible) parts using the visible parts after noise is added. Our method, however, is a consistency regularization method that focuses more on whether the noisy features lead to consistent outputs, rather than the ability to reconstruct them. Furthermore, we have experimented with the autoencoder module you mentioned, but did not observe any improvement. This may be because our model already has some denoising capability, and adding a decoder would introduce unnecessary parameters. Therefore, we did not include this module.
>
> [1] Kaiming He, Xinlei Chen, Saining Xie, Yanghao Li, Piotr Dollár,  and Ross Girshick. Masked Autoencoders Are Scalable Vision Learners[C]//Proceedings of the IEEE/CVF Conference on Computer Vision and Pattern Recognition. pp.16000-16009.2022.
>
> **W6:** It is unclear why the authors did not explore other model architectures.
>
> **Reply:** Thank you for your suggestion. We have added a comparison of results on another commonly used encoder, MLP, in Appendix A.8 (lines 940-1003), where we compare with different baselines. The results demonstrate that our approach also outperforms others on this encoder, showing that the proposed PGNA-PL model generalizes well across different backbones.
>
> **W7:** The experimental setup lacks details on hyperparameter tuning, such as the hyperparameter range and the optimized metric.
>
> **Reply:** In Appendix A.5 (lines 814-816), we have added the following clarification, especially regarding the hyperparameter range for $\beta_p$: ``We determined the optimal values of $\beta_p$ for various distributions across different datasets through a grid search method. The values of $\beta_p$ were taken from the range [0.5, 4], with an interval of 0.5 between points.''
>
> **W8:** The Leave-one-subject-out procedure is not implemented. This is important for generalization across unseen subjects rather than just trials from the same subjects.
>
> **Reply:** Cross-subject tasks for EEG-based emotion recognition are important, and we mention this in the Related Work section (line 698). It is also crucial to further explore scenarios where both PLL and cross-subject tasks occur simultaneously, as discussed in [1]. However, these two issues are fundamentally different. Given that the current application of PLL methods in EEG emotion recognition is limited, our model focuses on leveraging emotion relationships in the PLL scenario to disambiguate, as well as addressing the inherent low signal-to-noise ratio (SNR) in EEG, which could be effective across various EEG emotion recognition applications. Furthermore, in Appendix A.7 (lines 912-936), we demonstrate the advantages of our model under fully supervised conditions, further emphasizing the model’s potential contribution to related tasks. We plan to investigate joint solutions for PLL and cross-subject issues in future work.
>
> [1] Wei Li, Lingmin Fan, Shitong Shao, and Aiguo Song. Generalized contrastive partial label learning for cross-subject EEG-based emotion recognition. IEEE Transactions on Instrumentation and Measurement, 73, 2024.
>
> **W9:** The experiments are limited, especially in multi-task scenarios and naturalistic datasets. Simultaneous emotions and more diverse datasets should be considered.
>
> **Reply:** Please refer to the reply in W1.
>
> **W10:** The standard deviation in Tables 1–3 is very high, and macro/micro-F1 scores are reported without statistical significance tests. The effectiveness of the proposed approach cannot be justified against state-of-the-art methods.
>
> **Reply:** We have added the standard deviation data for both macro F1 and micro F1 scores in Tables 1-3. In most datasets and distributions, our method achieves optimal performance, and the standard deviations are also minimal or relatively low, which is particularly evident in the SEED and SEED-IV datasets. Additionally, in Appendix A.8 (lines 940-1003), we have included a comparison of results on another commonly used encoder, MLP, where we again find that our method achieves state-of-the-art performance in most cases. The experiments and data sufficiently demonstrate the model’s performance.

---

> ### Author Response · Authors · 2024-12-02
> **Response to Reviewer exz1 (Cont'd)**
>
> **W11:** The prototype-guided analysis visualizes latent relationships among emotions, but it is unclear how these relationships correlate with predictive performance.
>
> **Reply:** In the ablation study, as shown in lines 405-408, we have demonstrated the effectiveness of the prototype-guided module in improving performance. This module is based on the hypothesis that prototypes can learn inter-class relationships. To validate this hypothesis and provide interpretability for the module’s performance improvement, we conducted a visualization analysis in Section 3.2.3. Figures 3(a) and 3(b) show the inherent similarity between emotion labels under two generation methods, while Figures 3(c) and 3(d) visualize the inter-class distances of prototypes learned by the model during training. The high consistency between these results demonstrates that the prototypes can effectively learn the inter-class relationships we predefined when simulating candidate labels (as described in lines 485-525), allowing the model to guide the classifier’s disambiguation and emotion prediction capability through self-distillation.
>
> **Minor points:**
>
> **1:** The rationale for using only differential entropy features in EEG is unclear. The authors should provide an explanation for the feature extraction stage.
>
> **Reply:** The differential entropy feature of EEG has been proven to be a very effective frequency-domain feature [1,2]. Our research focuses on the application scenario of PLL and the signal-to-noise ratio of the EEG itself. The differential entropy feature is sufficient to support our study. Of course, raw EEG features may contain more noise, and using the raw EEG signal could be more beneficial for validating the effectiveness of our noise augmentation method. We will attempt this in future work.
>
> [1] Weilong Zheng and Baoliang Lu. Investigating critical frequency bands and channels for EEG-based emotion recognition with deep neural networks. *IEEE Transactions on Autonomous Mental Development*, 7(3):162–175, 2015.
>
> [2] Weilong Zheng, Wei Liu, Yifei Lu, Baoliang Lu, and Andrzej Cichocki. Emotionmeter: A multimodal framework for recognizing human emotions. *IEEE Transactions on Cybernetics*, 49(3):1110–1122, 2018.
>
> **2:** Notation of different equations is sometimes omitted or ill-defined, such as $v_{j}$ and $v_{i}$ in equation 1.
>
> **Reply:** We have thoroughly investigated and optimized this issue. For example, in lines 123-124, we have clarified the point more explicitly: "Let $v_i$ denote the GloVe vector for emotion category $e_i$, where $i$ indexes the emotion vocabulary."
>
> **3:** The overall organization of the paper needs improvement. For example, the feature preprocessing step to increase SNR in Section 3.3 should be placed before the methodological section. Moreover, some sentences disrupt the logical flow of the paper.
>
> **Reply:** The noise augmentation method proposed in this paper can also be considered a preprocessing method. However, we have improved the mixup process and adapted it to the PLL problem. Therefore, we have incorporated it into the Method section to highlight the innovation. Similarly, the classic MAE model [1] places the Mask process in the Approach section for description. Regarding the potential disruption of the logical flow of the paper, we have restructured the entire Method section (lines 108-313) and provided a detailed explanation of the logic behind the methods. We welcome further review on this matter.
>
> [1] Kaiming He, Xinlei Chen, Saining Xie, Yanghao Li, Piotr Dollár, and Ross Girshick. Masked Autoencoders Are Scalable Vision Learners[C]//Proceedings of the IEEE/CVF Conference on Computer Vision and Pattern Recognition. pp.16000-16009, 2022.
>
> **Questions:**
>
> **Q1:** The paper lacks a clear differentiation from existing partial label learning and self-distillation methods. How does your approach significantly differ from these methods?
>
> **Reply:** Please see the above response to W3 for details.
>
> **Q2:** Considering the complexity of emotions, why was GloVe chosen exclusively for semantic embedding? Are there alternative methods that might capture emotion multi-dimensionality more effectively?
>
> **Reply:** The reference to GloVe in this paper is intended to generate candidate labels by capturing the semantic relationships between emotion labels, which represents a novel exploration in the generation of candidate labels for emotion recognition. Approaches such as utilizing methods like BERT [1] to generate contextual embeddings are also worth further exploration, and we will investigate such techniques in future work.
>
> [1] Devlin J. BERT: Pre-training of deep bidirectional transformers for language understanding. *arXiv preprint arXiv:1810.04805*, 2018.

---

> ### Author Response · Authors · 2024-12-02
> **Response to Reviewer exz1 (Cont'd)**
>
> **Q3:** Could you elaborate on the hyperparameter tuning process, including the range of tested hyperparameters and optimization criteria?
>
> **Reply:** Please see the above response to W7 for details.
>
> **Q4:** Why was the leave-one-subject-out procedure not used, given its relevance for generalizability across unseen subjects in emotion recognition?
>
> **Reply:** Please see the above response to W8 for details.
>
> **Q5:** The experiments seem limited to specific tasks and datasets. How might the model perform across multiple emotional tasks and other naturalistic datasets?
>
> **Reply:** Please see the above response to W1 for details.
>
> **Q6:** Is the performance of the proposed approach significantly better than that of existing competitors?
>
> **Reply:** Please see the above response to W10 for details.

---

> ### Comment · Reviewer_exz1 · 2024-12-03
> **Response to authors' comment**
>
> The authors solved some of my primary concerns. This clarification helped to highlight the paper's novelty in the state-of-the-art context. As a result, I raised my score.

---

> > ### Author Response · Authors · 2024-12-03
> > **Thanks**
> >
> > Dear Reviewer exz1,
> >
> > Thank you for your detailed suggestions during the review process, as well as for your careful reading and thorough evaluation of our response. Your feedback has greatly helped us further improve the manuscript, particularly in clarifying the innovation of our research. Your recognition of this aspect is highly meaningful to us.
> >
> > We are deeply grateful for the time you spent reviewing our revisions and for the positive evaluation. Your acknowledgment of the innovation in our work has not only strengthened our confidence in this research direction but also inspired us to continue exploring this area further.
> >
> > Additionally, we would like to express our sincere appreciation for the increase in your score. Your support has provided significant encouragement for our research.
> >
> > Once again, thank you for your detailed review and invaluable feedback.
> >
> > Best Regards,
> >
> > The authors

---

### Official Review · Reviewer_5NVi · 2024-10-22

**Soundness:** 2
**Presentation:** 2
**Contribution:** 2
**Rating:** 5
**Confidence:** 5

**Summary:**

Summary of the paper:
The present paper proposes a novel approach for EEG-based emotion recognition. The authors propose a method that combines three things to enhance the accuracy of emotion predictions:
1) Semantic-based candidate label generation method for partial label learning (PLL) that incorporates the semantic relationships of emotions.
2) A model which uses emotional prototypes and self-distillation (utilizing KL divergence).
3) A novel noise augmentation approach for EEG data.

Through experiments using three publicly available emotion-based EEG datasets, the authors show that their proposed method achieves state-of-the-art emotion classification results.

**Strengths:**

Strengths:
1) Three novel proposals (PLL method + model using emotional prototypes + augmentation method), each of which might be useful for the EEG-based emotion recognition community.
2)	The proposed approach obtains good results consistently on three publicly available EEG-based emotion recognition datasets.
3)	It is well motivated that there are ambiguities and annotation errors when annotating emotion-related data.
4)	There is a simple and clever source of inspiration for developing a novel data augmentation strategy for EEG data.
5)	The algorithm listing is clear and concise.
6)	The majority of images and tables are constructed using vector graphics. In particular the tables are visually appealing and easy to read.

**Weaknesses:**

Weak points:
1) The authors make multiple strong statements and assumptions in the text which are not accompanied with strong evidence (literature/theoretical proof/strong empirical results). As an example, in the text, the authors claim that their proposed method “shows promising results”, “shows potential advantages”, “shows exceptional efficacy”, and shows “potential superiority” in recognizing phobias. If a model has a good classification accuracy of recognizing the emotional category "fear" (ranging between 0.64 to 0.75 in terms of recall), I would argue that one cannot deduce from this whether the model can be used to detect phobias. There are multiple reasons for this: First, phobias are very specific, intense fears which are triggered by particular situations or objects. Simply recognizing the emotional category “fear” well does not provide the context needed to identify a phobia. Second, to automatically detect phobias, machine-learning models would require training data that includes specific phobia-related scenarios and responses. Without data like this, the trained model might not generalize well for phobia detection. Third, phobias can be complex physiological and/or psychological responses, and a model trained for general-level emotion recognition might not be able to capture these kinds of nuances. Therefore, I would argue that the authors do not provide enough evidence to be able to state that their method shows “exceptional efficacy” in recognizing phobias.
2)	The motivation for the present work is severely lacking. As the motivation for using EEG-based emotion recognition, the authors state that “Traditional modes of emotion expression, such as facial expressions and spoken language, are easy to disguise, pushing the frontier towards emotion recognition through physiological signals as a more objective assessment method.”. However, EEG-based emotion recognition is far more invasive than video- or speech-based emotion recognition, and test subjects need to basically be in a lab or in lab-like conditions for EEG-based emotion recognition to be carried out. The authors do not give any use-cases or practical reasons on why EEG-based emotion recognition would be more convenient or applicable than using video- or speech-based emotion recognition.
3)	One of the key points of the proposed prototype-guided module is mentioned in lines 222-224: "As the iterations progress, different prototype features will provide stable representations for different emotion categories.". Without stableness, the entire PLL process proposed by the authors does not work. However, the authors do not provide anything to assure the reader that the stableness of their method is guaranteed, such as strong backing up using PLL literature, theoretical deductions/proof, or strong empirical results.
4)	There are countless typos in the text, making the reader feel like the manuscript was written in a hurry. Also, many concepts (e.g. the GloVe dictionary) are not well introduced to the reader, further adding to the impression that the manuscript is not well-polished.
5)	The text flows very unnaturally, in particular in the equations. As of now, the equations are not a natural part of the text, but they appear as if they were appendices within the text. I strongly suggest that the authors take a thorough look at how equations are typically presented in papers related to machine learning.
6)	Section 3 (Methods) is very difficult to follow, many parts and details are left unclear even after multiple read-throughs. It is essential that the description of the proposed novel approaches is clear for the reader.
7)	The present work would benefit from having notably more references. Currently, there are practically no occasions where there would be more than one reference to support the claims made by the authors.
8)	The majority of figure and table captions are not very descriptive, and these figures/tables cannot be understood completely without reading the main text first.
9)	Since the datasets that were used in the present study are public, it would be beneficial for both the authors and the EEG-based emotion recognition community to have the code publicly available. (Edit 3rd of December: This statement was false, as there was code provided in the supplementary material which I hadn't noticed.)
10)	It is not good practice to start sentences with a variable, e.g. line 200: "p is further processed by C..."
11)	It is also not very good practice to end a section with an equation, such as in Equation 8.
12)	I appreciate that the majority of images and tables are constructed using vector graphics. However, why are Figures 2 and 3 created with raster graphics? These figures would benefit from being produced in a vector graphics format.
13)	Even though the proceedings of the vast majority of conferences are nowadays not in printed format, it is still essential to consider how the paper appears if someone would print it in regular A4 format. For example, in Figures 2 and 3, there are multiple details that cannot be seen properly if the paper is printed in A4 format.
14)	Some of the colors of the visualizations are difficult to see against a white background, e.g. in Figure 3 parts (c) and (d).
15)	Table captions should always be above the tables (see Table 4).
16)	Even though ICLR accepts work on relevant application areas of machine learning such as  applications to neuroscience & cognitive science, in its current state the work is not interesting enough to bring value for the ICLR community. After revising the manuscript, I would strongly suggest that the authors submit the paper to some other relevant conference (e.g. IEEE EMBC 2025), where the target audience would be more suitable for the present study.

Initial recommendation: This paper should be rejected since it contains far too many weak points for it to be considered as an acceptable paper for ICLR (see list of weak points above). These weak points include, for example, multiple strong statements that are not backed up in any way, severely lacking motivation for conducting the present study, and poorly written methodological descriptions that are difficult to follow.

Here are more detailed notes:

Abstract:
1) The Abstract is a little bit confusing and hard to follow overall, having read it two times without reading the main text. However, after reading the main text through once, the Abstract was understandable. In particular, it was difficult to understand on the first read-through that there are three different things that are proposed: the label generation method, a model utilizing emotional prototypes and self-distillation, and a noise augmentation approach.
2)	The Abstract ends in a rather strong statement regarding the detection of phobias. Please see the “weak points” section above for a more detailed comment.


1. Introduction:
3) The first few sentences arouse the interest of the reader quite well. However, to support your strong claims (such as the strong linkage between negative emotions and conditions, and that EEG is notable for its precision and high temporal resolution), it would be highly beneficial to have more references. This same issue is repeated throughout the whole paper, where there are practically no occasions where there would be more than one reference to support the claims made by the authors.
4) Motivation is severely lacking: In its current state, EEG-based emotion recognition is a lot more difficult to set up and it is not even nearly as practical or convenient as recognizing emotions from facial (micro)expressions or spoken language. Therefore, the authors’ statement "pushing the frontier towards emotion recognition through physiological signals as a more objective assessment method" is not very valid, because many researchers working on emotion recognition from video or speech data are not progressing towards working on physiological signals; There is still plenty of work to be done in these fields alone, e.g. recognizing emotions from realistic, noisy input video data. Although, I do understand that physiological signal-based emotion recognition can be notably more accurate than recognizing emotions from e.g. videos if the person expressing the emotions is aiming to hide his/her emotions. I would like to see stronger motivation for when and where it would be beneficial or convenient to use an EEG-based emotion recognition setup over other setup types, "traditional modes of emotion expression are easy to disguise" is not a very strong motivation for the present work as it does not give any context on the actual use-cases of EEG-based emotion recognition.
5) "Among these, EEG is notable for its precision and high temporal resolution" --> EEG data is also notable for not generalizing well to machine-learning models, due to e.g. the large variability of EEG measurement setups and configurations (see e.g. https://www.sciencedirect.com/science/article/pii/S0925231224011251 and https://www.frontiersin.org/journals/human-neuroscience/articles/10.3389/fnhum.2021.765525/full).
6) It is well motivated that there are ambiguities and annotation errors when annotating emotion-related data. However, a few additional references would strengthen the given claims.
7) "The introduction of this method can significantly mitigate the impact of labeling errors in EEG-based emotion recognition tasks." --> References are needed to support this claim.
8) Line 080: “…of another samples…” --> “…of other samples…”


2. Related Work:
9) The Related Work section feels rushed and unfinished, and reading it leaves the reader a feeling that he/she has not been thoroughly guided through the work that is relevant to the field. There are many claims that are not given enough support through the use of references, e.g. lines 118-121 contain multiple claims, some of which are rather strong ones, but there are no references at all. The same thing applies for many other parts of the Related Work section, such as in lines 125-129. Additionally, this section is not coherent, for example the last paragraph begins with the sentence "Recent research suggests ABS methods may have greater potential.", greater potential in what? Since paragraphs should be independent entities that can be understood well on their own, this sentence is difficult to understand without a context.
10)	The explanation of generating emotionally semantic candidate labels is difficult to follow, in particular the notations for the equations could be clearer. Please see the “weak points” section above for a more detailed comment on the equation notation.
11)	Line 179: "...as shown in Equation 2. Where i represents the real emotion category." --> “...Equation 2, where i represents...”

3. Methods:

Section 3.2:

12) In the Introduction section, it is briefly mentioned what is the GloVe dictionary. However, you do not elaborate what GloVe is in the Methods section, which I consider to be a highly important piece of information for the reader to understand what the authors are trying to achieve in the present study. It should be explained to the reader what the GloVe dictionary is. Moreover, in the Abstract, Introduction, and Conclusion there is talk about the GloVe dictionary, in the Methods section there is talk about GloVe Vectors (and why is Vector with a capital letter?), and in the Experiments section there is talk about GloVe embeddings. I am assuming that all of these mean the same thing, so why are you using different terms in your text? Or if they mean different things, you should elaborate on this detail better.

13)	Lines 183-184: “This step ensures that the labels can be interpreted as probabilities, which is useful for training probabilistic models” --> Discriminative neural networks, such as CNNs that were used in the present study, are NOT probabilistic models! These models are deterministic, meaning that once trained, they give the same output for the same input. These models learn a mapping from the input to the output without modeling the probability distributions of the data.

Section 3.3:

14) When talking about differential entropy features, should there be a reference to https://ieeexplore.ieee.org/document/6695876?
15)	Line 195: section 3.2 --> Section 3.2
16)	The following sentence "n is the number of samples in the dataset, and m is the batch size." is completely disconnected. The variable m is not even used anywhere nearby in the text, so why is it defined in this sentence?
17)	Lines 197-198: "It is noted in the Figure 1 that basic modules contain an encoder E and a classifier C, which are used to extract high level features and achieve emotion recognition, respectively." This sentence has many issues. First of all, there isn't anything mentioned about Figure 1 beforehand, so "the" cannot be used before "Figure 1". Also, what is meant by a "basic module"? The entire text does not talk anything about basic modules before or after this appearance of the concept. How can the reader know what is meant by a "basic module"? Furthermore, when talking about classification models in neural networks, it is not very formal to say that these classifiers "achieve emotion recognition". A better way to phrase this would be e.g. that these classifiers "assigns the high-level features produced by the encoder to specific emotion categories" or "process the high-level features to determine the most likely emotion for each input signal" etc. etc.
18)	Line 198: "The function in E..." --> "The functions in E..."

Section 3.3.1:

19) Lines 222-224: "As the iterations progress, different prototype features will provide stable representations for different emotion categories." This is a very strong statement, that requires either strong backing up using PLL literature, theoretical deductions/proof, or strong empirical results.
20)	Line 227: "By calculating the distance-based similarity between the different prototype features". I am assuming that the authors should be referring to Equation 1?
21)	Lines 228-229: "As shown in Equation 7." --> This sentence is completely disconnected from the previous sentence.

Section 3.3.2:

22) Lines 244-246: "Considering that the differences in EEG signal distributions between different emotional categories are smaller than those between EEG signals and other types of noise..." This is a good source of inspiration for developing a novel data augmentation strategy for EEG data.
23)	Lines 247-249: "Unlike mixup, our method retains most of the current sample’s features without changing the corresponding candidate label." This is a strong claim that is backed up in the following text. However, in my opinion, it should be already briefly elaborated in this sentence that how does the proposed data augmentation method retain most of the current sample's features, as in its current state the sentence just raises more questions and confuses the reader. For example, "...without changing the corresponding candidate label by (something here). This is further detailed below."
24)	The algorithm listing is clear and concise.
25)	Equation 11: The formatting for x_i_noised should be further examined.
26)	Lines 262-264: "Furthermore, the Beta sampling method provided by Equation 9 offers a wider range of blending possibilities, enhancing the model’s generalization and robustness." --> This claim is not supported anywhere in the text, there are e.g. no experiments of references to show that this statement is valid.

Section 3.3.3:

27) Line 294: "...proposed by DNPL (Seo & Huh, 2021)" --> "...proposed by (Seo & Huh, 2021)". The method DNPL has not proposed anything, but the authors of the method have.

Section 3.3.4:

28) Why is there such a short section? This section should be merged to e.g. Section 3.3.3.
29)	Line 306: "L_overall" --> The appearance of this variable should be double-checked.

Section 3.3.5:

30) The title is misspelled.
31)	Lines 314-317: There are multiple errors/inaccuracies in the text. First of all, there isn't such a thing as a "batch normalization layer", it is just "batch normalization". Second, I believe the authors mean "...through a flattening operation..." and not "...through a flattening...". Third, it is just a single linear layer, not a "single-layer linear layer".

Section 4.1:

32) Why is "initial training set" in quotation marks?
33)	Why are evaluation metrics (e.g. Accuracy) written using capital letters?
34)	Line 342: "Appendix A.3", not "Appendices A.3".
35)	Line 347: It should be mentioned what the Russell distribution is. Now it is only briefly mentioned in the Introduction section, and there it is not very well detailed either.

Section 4.2.1:

36) Line 355: Why is Macro written with a capital letter?
37)	Line 365: The authors mention potentially unstable training as a drawback for IBS-based PLL methods. However, they do not elaborate anywhere in the manuscript on how their approach produces a stable training process.
38)	Line 404: Why is Semantic written with a capital letter?

Section 4.2.2:

39) Line 411: Again, why is Semantic written using capital letters?
40)	Figure 2: The datasets and conditions should be visualized in the subplots. Now it is not explicitly shown (or elaborated on in the caption) that which of the subplots correspond to which dataset or label distribution.
41)	If a model has a good classification accuracy of recognizing the emotional category "fear" (approx. 70%), I would argue that one cannot deduce from this whether the model can be used to detect phobias. Please see the “weak points” section above for a more detailed comment.
42)	In Section 4.1, the authors refer to the emotional categories using lower case letters, but in this section the authors use capital letters. The same applies for Section 4.2.3.

Section 4.2.3:

43) It is not explained well how PCA was used for the visualization of Subfigures (c) and (d) of Figure 3. From which features were the PCA features computed? For visualizing that data in 2D, how many principal components were used in the calculation of PCA? And were the two highest-variance components then selected for the visualization to get a 2D image? Is there only one prototype for each emotion, or are the shown categories somehow fused together from the prototypes of the dataset samples?
Conclusion:
44)	Concise wrap-up to the present work.
45)	Line 539: “The confusion matrix suggests our method’s potential superiority in treating phobias.” --> This is a very strong statement, please see the “weak points” section above for a more detailed comment.

References:

46) There are a little over 30 references. The text would most certainly benefit from a stronger linkage to theory and prior work through the use of more references.
47)	The bibliography style is not consistent: Some references have page numbering using the abbreviation “pp.” while others don’t, some journals have ISSN and/or DOI displayed and some don’t etc. etc.
48)	Some title names need to be reformatted in terms of capital letters. For example, the abbreviation for principal component analysis never appears in the form “pca”, but it is always in capital letters, i.e. “PCA”. As another example, electroencephalography is typically abbreviated with capital letters, i.e. EEG, but now some appearances of the abbreviation are with small letters (eeg) and others have the same abbreviation with capital letters (EEG).

Appendix A.1:

49) Line 647: "Conv Layer" is not a proper way of writing "convolutional layer" in the body text.

Appendix A.2:

50) "...focused on binary or three emotion classification." --> "...focused on binary or ternary emotion classification."
51)	Line 675: "The SEED-IV dataset extends the SEED dataset..." --> It is not elaborated whether the SEED dataset is included in SEED-IV (as the term "extends" suggests), or if it is a completely distinct dataset. Given that there are additional emotion categories, the latter option seems more probable.
52)	Line 683: Why is "Disgust" written with a capital letter?
53)	Lines 690-691: "To prepare for model training, we normalized the vector of each DE feature as input." --> What normalization strategy what used? For example, was each 4-second segment normalized at segment-level, or was each 4-second segment normalized at dataset-level? Also, was the normalization carried out so that there would be zero-mean and unit-variance segments, or was some other normalization method used?
54)	It is not explicitly mentioned how many samples are there in each of the three datasets. This information is very important in machine learning, as it impacts many details regarding the model training process.

Appendix A.3:

55) Line 701: Why are the random seed numbers explicitly mentioned? This information is not useful at all.

Appendix A.4:

56) Line 733: "Notably effective in EEG-based emotion recognition applications." --> This is not a well-formulated sentence.
57)	Lines 735-754: The sentences are not well-formulated. For example, the sentence on line 745 should start something like "The method performs..." etc., and not "Performs...".
58)	It is very difficult to understand how the presented ABS and IBS methods work, since the methods are presented very shortly but with lots of technical details that would need a larger context. For example, on line 742 it is mentioned that "...uses intrinsic representations learned through a CAV model...", but nowhere in the entire manuscript there is anything said about what these CAV models are, or let alone what the abbreviation CAV even stands for.


Edit 3rd of December: Raising score from 1 --> 5

**Questions:**

See detailed notes.

---

> ### Author Response · Authors · 2024-12-02
> **Response to Reviewer 5NVi**
>
> ## **Weaknesses:**
>
> **W1:** The authors make strong claims, such as "exceptional efficacy" in recognizing phobias, but do not provide sufficient evidence, particularly regarding the specific context needed for phobia detection.
>
> **Reply:** Thank you for pointing this out. We have adopted your suggestion and revised the statement regarding the recognition of phobias. It now mentions the potential help in recognizing fear-related disorders, which is more appropriate given the high accuracy of our method in identifying fear. As stated in the last sentence of the abstract (lines 26-28): "Our method effectively disambiguates complex emotions and shows promising results in assisting in the recognition of fear-related disorders."
>
> **W2:** The motivation for using EEG-based emotion recognition is unclear. The authors do not provide practical use-cases or reasons why EEG is more applicable than video or speech-based methods.
>
> **Reply:** We have optimized the sentence you mentioned, as seen in lines 34-45: "Physiological signals, due to their difficult-to-disguise nature, are suitable for the objective assessment of emotional responses." Additionally, we have provided three citations supporting similar claims after this sentence, transforming the statement into a more cautious one, supported by relevant references.
>
> **W3:** The authors do not provide evidence (literature, theory, or empirical results) to ensure its stability of the prototype-guided module during the learning process.
>
> **Reply:** To mitigate the risks associated with stability claims, we have followed your suggestion and removed this part. Instead, we emphasize the prototype’s ability to learn inter-class relationships. As stated in lines 208-210, learning these relationships also helps the model disambiguate different labels, which is further validated by our ablation study.
>
> **W4:** There are numerous typos in the manuscript, and some concepts (e.g., GloVe dictionary) are not well introduced, giving the impression that the manuscript is hastily written.
>
> **Reply:** Thank you for your thorough review. We have read through the entire manuscript again and made the corrections for typos, as per your notes. Furthermore, we have followed your suggestion and added descriptions of GloVe and the Russell Distribution in Appendix A.2 (lines 730-759).
>
> **W5:** The flow of the text, especially around equations, is unnatural. The equations appear more like appendices within the text rather than integral parts.
>
> **Reply:** We have revised the entire Methods section (lines 108-313) based on your notes, with a particular focus on improving the descriptions and explanations of the formulas. We believe these changes have significantly enhanced the clarity of the section, and we invite you to review them further.
>
> **W6:** Section 3 (Methods) is difficult to follow, with unclear details even after multiple readings. The description of the methods should be clearer.
>
> **Reply:** We have provided more detailed descriptions of the methods mentioned in the Methods section, enhancing the clarity and granularity of the explanations.
>
> **W7:** The manuscript would benefit from more references to support the authors' claims.
>
> **Reply:** Following your suggestion, we have added more citations to support the relevant claims. The number of references has increased from 30 to 47. Specifically, we have improved some of the statements that were previously unreasonable, such as those related to the direct detection of phobias.
>
> **W8:** The majority of figure and table captions are not very descriptive, and these figures/tables cannot be understood completely without reading the main text first.
>
> **Reply:** As per item 40 in your notes, we have added captions for the six subplots in Figure 2 (lines 452-453, lines 465-466) to help readers understand and differentiate the meanings of the subplots.
>
> **W9:** As the datasets are public, the authors should consider making the code available to benefit both their work and the EEG-based emotion recognition community.
>
> **Reply:** We have uploaded the code in the Supplementary Material, which is reproducible and complete. We plan to further upload it to GitHub upon the paper being published.
>
> **W10:** It is not good practice to start sentences with a variable, e.g., line 200: "$p$ is further processed by $C$..."
>
> **Reply:** We have thoroughly examined the issue and made comprehensive optimizations. The sentence you referred to now corresponds to line 181 in the restructured version: "This high-level feature is then fed into the classifier to obtain the predicted logits $\hat{y}_i$."

---

> ### Author Response · Authors · 2024-12-02
> **Response to Reviewer 5NVi (Cont'd)**
>
> **W11:** It is also not very good practice to end a section with an equation, such as in Equation 8.
>
> **Reply:** We have optimized this type of issue, such as adding the meaning of the loss function under Equation 8 and summarizing Section 2.3.1 (lines 227-235). Similarly, we also provided a summary at the end of Section 2.3.2 (lines 281-285). These changes have made the structure more clear and compact.
>
> **W12:** Figures 2 and 3 use raster graphics, which should be replaced with vector graphics for better quality.
>
> **Reply:** Thank you for pointing out this issue. Figures 2 and 3 were generated directly as PNG images using matplotlib, and this may have affected their quality when imported into LaTeX. We apologize for not noticing this earlier and for not optimizing the images before the submission deadline. However, we promise that before the paper is published, all images will be converted to PDF format, resolving this issue entirely.
>
> **W13:** The paper should consider how it would appear when printed in A4 format, as some details in Figures 2 and 3 are hard to see when printed.
>
> **Reply:** Thank you for your suggestion. We have increased the font size of the words on the axes in Figures 2 and 3, and tilted the labels on the x-axis to prevent overlap. Additionally, the numbers and words in Figures 3(c) and 3(d) have been enlarged. As mentioned in W12, we will convert these images to PDF format in the future, which is expected to further improve their quality.
>
> **W14:** Some of the colors in the visualizations are hard to see against a white background (e.g., in Figure 3 parts (c) and (d)).
>
> **Reply:** Thank you for your suggestion. We have applied a light gray background to Figures 3(c) and 3(d) (lines 499-507), making the colors in the figures clearer.
>
> **W15:** Table captions should always be above the tables (see Table 4).
>
> **Reply:** Thank you for pointing out this typo. We have made the corrections to lines 1019, 1026, and 1036.
>
> **W16:** The paper may not be of sufficient interest to the ICLR community. It might be better suited for a conference like IEEE EMBC 2025, where the target audience aligns more closely with the study.
>
> **Reply:** Thank you for your suggestion. However, the PLL method is a classical weakly supervised learning approach, and works like PiCO[1] have already been published in ICLR and are widely recognized. In recent years, related work on EEG-based emotion recognition has also been accepted at ICLR, such as [2]. In our work, the application of prototypes is an approach that addresses the representation learning problem in weakly supervised settings, and thus fits within the scope of ICLR.
>
> [1] Haobo Wang, Ruixuan Xiao, Yixuan Li, Lei Feng, Gang Niu, Gang Chen, and Junbo Zhao. Pico: Contrastive label disambiguation for partial label learning. In International Conference on Learning Representations, 2022.
>
> [2] Chenyu Liu, Xinliang Zhou, Zhengri Zhu, Liming Zhai, Ziyu Jia, and Yang Liu. VBH-GNN: Variational Bayesian Heterogeneous Graph Neural Networks for Cross-subject Emotion Recognition[C]//The Twelfth International Conference on Learning Representations, 2024.
>
> ## **Detailed Notes:**
>
> ### **Abstract:**
>
> **1:** The Abstract is unclear, particularly regarding the three proposed methods (label generation, prototype-based model, noise augmentation). Clarification is needed to improve readability.
>
> **Reply:** We have rewritten the Abstract, with a particular emphasis on the problems addressed by the three innovations we propose, as outlined in lines 15-23.
>
> **2:** The Abstract ends in a rather strong statement regarding the detection of phobias.
>
> **Reply:** The issue has been optimized, and it has been revised to: "promising results in helping recognize fear-related disorders."
>
> ### **Introduction:**
>
> **3:** More references are needed to support claims regarding the strong linkage between negative emotions and conditions, and EEG's precision and high temporal resolution.
>
> **Reply:** We have added some references in the introduction to support the relevant claims, such as the sentence in lines 39-40: "EEG, due to its precision and high temporal resolution, has already been extensively studied," with three references provided.
>
> **4:** The motivation for EEG-based emotion recognition is weak. More context is needed on its practical benefits compared to facial expressions or speech-based recognition.
>
> **Reply:** We have toned down this claim, as seen in lines 34-35, and now only mention: "Physiological signals, due to their difficult-to-disguise nature, are suitable for the objective assessment of emotional responses." supported by relevant references.

---

> ### Author Response · Authors · 2024-12-02
> **Response to Reviewer 5NVi (Cont'd)**
>
> **5:** EEG is noted for precision, but it also suffers from poor generalization to machine learning models due to large variability in measurement setups. More references are needed to support this point.
>
> **Reply:** Here, we intended to mention the advantages of EEG to introduce the reason for its use in emotion recognition. However, the point you raised concerns the challenges faced by EEG. To avoid ambiguity, we have revised this to: "Among these, EEG, due to its precision and high temporal resolution, has already been extensively studied" as stated in lines 38-40, and the sentence is supported by relevant references.
>
> **6:** Additional references are needed to strengthen the claim about ambiguities and annotation errors in emotion-related data.
>
> **Reply:** We have added the following reference in line 45:
>
> Wei-Bang Jiang, Yu-Ting Lan, and Bao-Liang Lu. *REmoNet: Reducing emotional label noise via multi-regularized self-supervision*. In *Proceedings of the 32nd ACM International Conference on Multimedia*, pp. 2204–2213, 2024.
>
> This reference explicitly addresses the issue of emotional label noise in EEG signals.
>
> **7:** Like the sentence "The introduction of this method can significantly mitigate the impact of labeling errors in EEG-based emotion recognition tasks.", references are needed to support this claim.
>
> **Reply:** The citation below has been added in lines 55-56 to support this claim:
>
> Guangyi Zhang and Ali Etemad. *Partial label learning for emotion recognition from EEG*. arXiv preprint arXiv:2302.13170, 2023.
>
> **8:** Line 080: "…of another samples…" should be changed to "…of other samples…"
>
> **Reply:** The suggestion has been adopted and modifications have been made in line 81.
>
> ### **Related Work:**
>
> **9:** The Related Work section feels rushed and lacks sufficient references (e.g., lines 118-121, 125-129). Additionally, the section is incoherent, e.g., the sentence "Recent research suggests ABS methods may have greater potential" is vague without context.
>
> **Reply:** We have reorganized the logic of the related work. Due to the length of the content, we moved it to Appendix A.1 (lines 695-728) and added appropriate citations. Additionally, upon further review, we found that our method and the DNPL method do not belong to the previously mentioned ABS method. As a result, we no longer mention the irrelevant ABS method and introduce DNPL as a special dynamic label probability adjustment method (lines 714-718).
>
> **10:** The explanation of generating emotionally semantic candidate labels is unclear, particularly regarding the equation notations.
>
> **Reply:** We have adopted your suggestion and made overall improvements to the description and formula in section 2.2 (lines 119-150). The optimized content now provides a clearer explanation of the steps in our proposed candidate label generation method.
>
> **11:** Line 179: "…as shown in Equation 2. Where i represents the real emotion category." should be changed to "…Equation 2, where i represents…"
>
> **Reply:** Thank you for pointing out the typo. The issue has been resolved in the revised statement, as seen in lines 132-133.
>
> ### **Methods:**
>
> #### **Section 3.2:**
>
> **12:** The GloVe dictionary is mentioned but not elaborated on in the Methods section. The terms "GloVe dictionary", "GloVe Vectors" (and why is Vector with a capital letter?) and "GloVe embeddings" are inconsistent and should be clarified.
>
> **Reply:** We have added an introduction to the GloVe dictionary in Appendix A.2.1 (lines 732-742). The word "vectors" in "GloVe vectors" has been changed to lowercase. We have standardized the terminology to align with the official GloVe website, using "vectors" consistently instead of dictionary, vectors, or embeddings.
>
> **13:** Line 183-184: The statement about probabilistic models is incorrect. CNNs are deterministic models, not probabilistic.
>
> **Reply:** We have accepted this suggestion and removed the sentence.
>
> #### **Section 3.3:**
>
> **14:** When talking about differential entropy features, should there be a reference to https://ieeexplore.ieee.org/document/6695876?
>
> **Reply:** The citation has been added in line 154.
>
> **15:** Line 195: "section 3.2" should be corrected to "Section 3.2".
>
> **Reply:** The content has been optimized. Due to structural adjustments (Related work moved to Appendix A1), the revised content is now in Section 2.2 as seen in line 158.
>
> **16:** The sentence "n is the number of samples in the dataset, and m is the batch size" is disconnected as "m" is not used later in the text.
>
> **Reply:** We have moved the definition of the variable $m$ closer to equations (8) and (12), as seen in lines 227-228 and lines 300-301, to better support the descriptions of the relevant formulas.

---

> ### Author Response · Authors · 2024-12-02
> **Response to Reviewer 5NVi (Cont'd)**
>
> **17:** Line 197-198: The sentence about Figure 1 is unclear. The term "basic module" is introduced without prior context, and the phrasing of "achieve emotion recognition" is too informal. A better way to phrase this would be e.g. that these classifiers "assigns the high-level features produced by the encoder to specific emotion categories" etc.
>
> **Reply:** Your suggestion has been adopted: we have removed "the" before Figure 1 and no longer mention "basic modules." Additionally, regarding the usage of the classifier, we have incorporated your recommendation. The relevant changes can be found in lines 159-161: "As illustrated in Figure 1, the PGNA-PL model comprises an encoder $E$ and a classifier $C$, parameterized by $\theta$ and $\delta$, respectively. The encoder $E$ extracts high-level features from the input data, and the classifier $C$ assigns these features to specific emotion categories."
>
> **18:** Line 198: Correct "The function in E..." to "The functions in E..."
>
> **Reply:** The suggestion has been adopted and optimized, as seen in line 161: "The functions of $E$ and $C$ are denoted by $f_{\theta}$ and $F_{\delta}$."
>
> ### Section 3.3.1:
>
> **19:** Line 222-224: The statement on stable representations for emotion categories needs stronger backing via PLL literature, theoretical proof, or empirical results.
>
> **Reply:** We have removed the phrase "stable representations" and replaced it with a description focused on learning inter-class distances, as in lines 208-210: "Over time, these prototypes represent the various emotion categories, and the distances between them capture the relationships between different emotions. As a result, the classifier can more easily distinguish between similar emotions based on the evolving prototypes."
>
> **20:** Line 227: "By calculating the distance-based similarity between the different prototype features". I am assuming that the authors should be referring to Equation 1?
>
> **Reply:** This section describes equation (7). To clarify further, we have optimized the related content in lines 212-216.
>
> **21:** Line 228-229: "As shown in Equation 7" seems disconnected from the previous sentence.
>
> **Reply:** We have adopted this suggestion and removed the sentence. Currently, a detailed description of equation (7) is provided in lines 212-216.
>
> ### Section 3.3.2:
>
> **22:** Line 244-246: "Considering that the differences in EEG signal distributions between different emotional categories are smaller than those between EEG signals and other types of noise..." This is a good source of inspiration for developing a novel data augmentation strategy for EEG data.
>
> **Reply:** Thank you for your appreciation of this point. Unfortunately, during a recent revision, we inadvertently deleted this sentence. We apologize for this oversight and will ensure it is reinstated before the paper is formally published.
>
> **23:** Line 247-249: The claim about mixup vs. the proposed method requires more elaboration on how the current sample's features are retained.
>
> **Reply:** To avoid confusion from previous wording, we have revised this content in lines 249-251, focusing on the differences between our proposed noise-adding method and the mixup data augmentation approach: "However, unlike traditional mixup, our approach focuses on adding noise by blending two EEG signals, thus maintaining the structure of the signal while introducing perturbations that simulate noise." And lines 259-262 further explain how to preserve the characteristics of the original EEG samples in detail.
>
> **24:** The algorithm listing is clear and concise.
>
> **Reply:** Thank you for your appreciation of this section. We have also further consolidated unnecessary line breaks to make the algorithm list more compact, as seen in lines 325-326 and lines 328-329.
>
> **25:** Equation 11: The formatting for $x_{i\_noised}$ should be further examined.
>
> **Reply:** As in equation (1) at line 266, this has been optimized to $x_i'$.
>
> **26:** Line 262-264: The claim about Beta sampling and model robustness lacks supporting evidence or experiments.
>
> **Reply:** We have removed the related statement and provided a more detailed description of the impact of the hyperparameter ($\beta_p$) in lines 253-256.
>
> ### Section 3.3.3:
>
> **27:** Line 294: Correct citation for DNPL method—refer to authors, not the method itself.
>
> **Reply:** As seen in line 289, we have optimized this to "We use the naive loss function from DNPL."
>
> ### Section 3.3.4:
>
> **28:** Why is there such a short section? This section should be merged to e.g. Section 3.3.3.
>
> **Reply:** We have adopted this suggestion and integrated the descriptions of the classification loss and overall loss into Section 2.3.3, as found in lines 287-305.
>
> **29:** Line 306: Double-check appearance of $L_{overall}$.
>
> **Reply:** This typo has been corrected, as seen in line 302, where the term is now expressed as $L_{overall}$.

---

> ### Author Response · Authors · 2024-12-02
> **Response to Reviewer 5NVi (Cont'd)**
>
> ### Section 3.3.5:
>
> **30:** The title is misspelled.
>
> **Reply:** This typo has also been corrected. As seen in line 307, it is now stated as "ARCHITECTURE DETAILS."
>
>
> **31:** Lines 314-317: There are multiple errors/inaccuracies in the text. First of all, there isn't such a thing as a "batch normalization layer", it is just "batch normalization". Second, I believe the authors mean "...through a flattening operation..." and not "...through a flattening...". Third, it is just a single linear layer, not a "single-layer linear layer".
>
> **Reply:** The relevant optimization has been completed. As stated in lines 309-313: "The encoder consists of two CNN modules; each includes a 1-D convolutional layer, batch normalization, and a LeakyReLU activation function. The output is then transformed to the desired dimensions $d$ through a flattening operation and a linear layer. The classifier comprises a single linear layer designed for emotion recognition."
>
> ### Section 4.1:
>
> **32:** Why is "initial training set" in quotation marks?
>
> **Reply:** We directly replaced the initial training set with the training set and then described the process of splitting the validation set in line 352: "We randomly select 20% of the training set to serve as the validation dataset, with the remaining data used for training the model."
>
> **33:** Why are evaluation metrics (e.g. Accuracy) written using capital letters?
>
> **Reply:** This has been changed to lowercase, as shown in lines 353-354.
>
> **34:** Line 342: Should be "Appendix A.3," not "Appendices A.3."
>
> **Reply:** This has been adopted. As shown in line 355, we have rearranged the Appendix and updated it to "the hyperparameters can be found in Appendix A.5." Additionally, we reviewed and corrected similar issues throughout the paper.
>
> **35:** Line 347: Clarify what the Russell distribution is, as it’s only briefly mentioned in the Introduction.
>
> **Reply:** We have provided a more detailed description of this section in Appendix A.2.2 (lines 744-758), and a hyperlink to this Appendix has been included in line 87.
>
> ### Section 4.2:
>
> #### Section 4.2.1:
>
> **36:** Line 355: Why is Macro written with a capital letter?
>
> **Reply:** The terms "Macro" and "Micro" have been changed to lowercase, as shown in line 354.
>
> **37:** The authors mention unstable training for IBS-based PLL methods but do not explain how their approach ensures stable training.
>
> **Reply:** As shown in lines 193-196, the IBS methods, including PiCO, update subsequent labels in an unchanged manner, causing the optimization target to continuously vary, which can lead to instability. Our method, based on DNPL, optimizes the model's disambiguation ability with a fixed learning target, thus avoiding such instability.
>
> **38:** Line 404: Why is Semantic written with a capital letter?
>
> **Reply:** The complete Semantic Distribution and Russell Distribution are now capitalized to define two distinct terms frequently used in the paper, rather than just capitalizing the first letter of "Semantic" and "Russell."
>
> #### Section 4.2.2:
>
> **39:** Line 411: Again, why is Semantic written using capital letters?
>
> **Reply:** As shown in reply of number 38.
>
> **40:** The datasets and conditions in Figure 2 should be labeled in the subplots or caption for clarity.
>
> **Reply:** This has been added in the figure legends, as shown in lines 452-453 and 462-463.
>
> **41:** A model with 70% accuracy in recognizing "fear" does not necessarily indicate its ability to detect phobias. Refer to the "weak points" section for more detail.
>
> **Reply:** This has been optimized, as shown in line 431: "in assisting in the recognition of fear-related disorders."
>
> **42:** The authors use both lowercase and capitalized letters for emotional categories in Sections 4.1 and 4.2.3; consistency is needed.
>
> **Reply:** We have standardized all emotion category terms to lowercase, as shown in line 430.
>
> ### Section 4.2.3:
>
> **43:** PCA usage for visualizing Subfigures (c) and (d) in Figure 3 is unclear. Details about feature selection and principal components need to be provided.
>
> **Reply:** The PCA features come from the prototype features obtained at the end of training. Since cross-validation generates multiple prototypes, we averaged the prototype features for different emotions and reduced the prototypes from the original dimension to two principal components. Thus, all principal components in the data were used in PCA, but only the first two components with the highest variance were selected in the final results.
>
> **44:** Concise wrap-up to the present work.
>
> **Reply:** Thank you for your appreciation.
>
> **45:** Line 539: "The confusion matrix suggests our method’s potential superiority in treating phobias." is a strong statement and needs more evidence. See "weak points" section.
>
> **Reply:** This has been optimized, as shown in line 539: "in helping to identify fear-related disorders."

---

> ### Author Response · Authors · 2024-12-02
> **Response to Reviewer 5NVi (Cont'd)**
>
> ### References:
>
> **46:** There are a little over 30 references. The paper would benefit from more references to strengthen the connection to theory and prior work.
>
> **Reply:** Thank you for your suggestion. We have increased the number of references to 47.
>
> **47:** The bibliography style is inconsistent (e.g., the use of "pp.", ISSN/DOI, etc.).
>
> **Reply:** We have standardized the formatting of the references. The use of "pp." when it appears is defined by the conference's provided template. We found that journals generally do not use "pp.", while conferences and books do. We have temporarily removed the two additional DOI information to ensure consistency. If the organizing committee requests us to add them later, we will provide the relevant details as needed.
>
> **48:** Title formatting needs adjustment (e.g., "PCA" and "EEG" should consistently use capital letters).
>
> **Reply:** We have thoroughly checked and resolved this issue. All words that are conventionally capitalized, including EEG, PCA, and proper nouns specified by the authors in titles, have been capitalized.
>
> ### Appendix A.1:
>
> **49:** Line 647: "Conv Layer" is not a proper way of writing "convolutional layer" in the body text.
>
> **Reply:** As seen in line 762, the convolutional layer has been optimized.
>
> ### Appendix A.2:
>
> **50:** "binary or three emotion classification" should be changed to "binary or ternary emotion classification."
>
> **Reply:** This has been adopted, as seen in line 785, where ternary emotion classification has been optimized.
>
> **51:** Line 675: Clarify if the SEED dataset is included in SEED-IV or if it is distinct, given that SEED-IV adds emotion categories.
>
> **Reply:** Yes, it is a completely different dataset. As seen in line 788, we have updated this to: "The SEED-IV dataset introduces four distinct emotions."
>
> **52:** Line 683: Why is "Disgust" written with a capital letter?
>
> **Reply:** As mentioned in the response to issue 42 above, we have standardized all emotion words to lowercase. This optimization is reflected in line 796.
>
> **53:** Lines 690-691: Clarify the normalization strategy for DE feature vectors—was it segment-level or dataset-level normalization? Was zero-mean and unit-variance normalization used?
>
> **Reply:** We use the Min-Max normalization method, applied to the entire dataset, which scales all feature values to a given range between the minimum and maximum values. In this paper, the minimum and maximum values are set to 0 and 1, respectively. This method does not consider the mean or variance of the data. We have included this information in Appendix A.4 (lines 803-809).
>
> **54:** The number of samples in each dataset should be specified, as this impacts model training details.
>
> **Reply:** We reviewed the relevant literature and found that in this field, particularly in papers using these datasets, statistical analysis of such data is rarely performed. Moreover, we have detailed the experimental setup in the paper and uploaded a one-click runnable code, so there should be no ambiguity regarding how the dataset is used.
>
> ### Appendix A.3:
>
> **55:** Line 701: The explicit mention of random seed numbers is unnecessary.
>
> **Reply:** These specific numbers have been removed.
>
> ### Appendix A.4:
>
> **56:** Line 733: The sentence "Notably effective in EEG-based emotion recognition applications" is poorly formulated.
>
> **Reply:** The sentence has been removed.
>
> **57:** Lines 735-754: The sentences are not well-formulated. For example, the sentence on line 745 should start something like "The method performs..." etc., and not "Performs...".
>
> **Reply:** We have revised the beginning of most baseline method descriptions to include the model's name (except for CR, as this name is not officially designated), resolving this issue. This can be seen in lines 847-910.
>
> **58:** The presentation of the ABS and IBS methods is unclear due to a lack of context, especially regarding the CAV model mentioned on line 742, which is not explained.
>
> **Reply:** We have expanded the details of different baselines in Appendix A.6 to help readers understand, as seen in lines 847-910. This includes an introduction to CAV (lines 861-863). Additionally, after careful review, we found that DNPL in the baseline is not part of the ABS method, since our method is based on DNPL and thus does not belong to ABS. Therefore, we have standardized the removal of all references to ABS.

---

> > ### Comment · Reviewer_5NVi · 2024-12-03
> > **Response to authors**
> >
> > Thank you for the very thorough revision of the paper and the detailed responses to each of my comments, I highly appreciate it. I will raise my score.

---

> > > ### Author Response · Authors · 2024-12-03
> > > **Thanks**
> > >
> > > Dear Reviewer 5NVi,
> > >
> > >
> > > Thank you very much for your thoughtful and constructive feedback during the review process. Your comments were invaluable in helping us improve the clarity and quality of our paper. We carefully addressed each of your suggestions, which have significantly strengthened our work. We are grateful for your positive evaluation of the revised manuscript.
> > >
> > >
> > > Your feedback has not only enhanced the paper, but it will also inspire us to continue exploring further in this area of research. Your insights have been a great source of motivation for our future work.
> > >
> > >
> > > Once again, thank you for your time, effort, and support.
> > >
> > >
> > > Best regards,
> > >
> > > The authors

---

> > > > ### Comment · Reviewer_5NVi · 2024-12-03
> > > > **Response to authors' comment**
> > > >
> > > > Thank you for the kind feedback! Also, it is always great to hear that the comments were useful.

---

> > ### Comment · Reviewer_5NVi · 2024-12-03
> > **Response to authors #2**
> >
> > Oh, and regarding weakness #9 I listed:
> > I apologize for not noticing that there was code provided in the supplementary material.

---

### Official Review · Reviewer_9xuQ · 2024-10-29

**Soundness:** 3
**Presentation:** 2
**Contribution:** 2
**Rating:** 5
**Confidence:** 3

**Summary:**

This paper, EEG-Based Emotion Recognition via Prototype-Guided Disambiguation and Noise Augmentation in Partial-Label Learning, presents the PGNA-PL model, leveraging Partial Label Learning (PLL) to enhance emotion recognition from EEG signals. Recognizing challenges like noisy data and ambiguous labels in traditional emotion recognition, the authors use a semantic-based candidate label generation method with GloVe embeddings to capture relationships between overlapping emotions. The PGNA-PL model incorporates a prototype-guided noise augmentation module that improves classification accuracy by disambiguating labels and enhancing robustness through controllable noise injection. Evaluations on SEED, SEED-IV, and SEED-V datasets demonstrate state-of-the-art (SOTA) performance, especially in detecting nuanced emotional states.

**Strengths:**

1) Innovative Use of Semantic Embeddings: The GloVe-based candidate label generation effectively captures the relationships between overlapping emotions, improving the model's ability to distinguish nuanced emotional states.
2) Enhanced Robustness with Noise Augmentation: The PGNA-PL model’s noise augmentation strategy, inspired by the mixup method, adds controlled noise to enhance resilience against EEG’s low SNR.
3) Comprehensive Evaluation and Ablation Studies: Evaluations on three major datasets, including SEED, SEED-IV, and SEED-V, provide a robust validation of the model’s effectiveness.

**Weaknesses:**

1) Dependency on GloVe Embeddings: The reliance on GloVe embeddings may limit adaptability, particularly as other, more dynamic contextual embeddings (e.g., from transformer models) could provide richer semantic information for label disambiguation.
2) Potential Instability with Prototype Guidance: While the prototype guidance module stabilizes the model by not updating candidate labels, there could still be some instability, especially in EEG settings where signals are prone to noise.
3) Noise Augmentation Choices: The study could examine how different types of noise augmentation affect model robustness. Currently, the mixup method is effective, but real-world noise may be more complex.

**Questions:**

1) How does the model handle datasets  like DEAP?
2) What are the implications of using alternative embeddings?
3) How does the model address unseen emotions or emotions not mapped in the initial GloVe embedding setup?

---

> ### Author Response · Authors · 2024-12-02
> **Response to Reviewer 9xuQ**
>
> **Weaknesses:**
>
> **W1:** Reliance on GloVe embeddings may limit adaptability; dynamic contextual embeddings like transformer models could offer richer semantic information.
>
> **Reply:** The reference to GloVe in this paper is intended to generate candidate labels by capturing the semantic relationships between emotion labels, which represents a novel exploration in the generation of candidate labels for emotion recognition. The approach you suggested, such as utilizing methods like BERT [1] to generate contextual embeddings, is also worth further exploration and we will investigate such techniques in future work.
>
> [1] Devlin J. BERT: Pre-training of deep bidirectional transformers for language understanding [J]. arXiv preprint arXiv:1810.04805, 2018.
>
> **W2:** While the prototype guidance module stabilizes the model by not updating candidate labels, there could still be some instability, especially in EEG settings where signals are prone to noise.
>
> **Reply:** We agree with your point, and have removed the description about prototype-guided enhancement of stability from the paper. Instead, we emphasize the classification benefits brought by the learning of inter-class distances through prototypes. As stated in lines 208-210: "Over time, these prototypes represent the various emotion categories, and the distances between them capture the relationships between different emotions. As a result, the classifier can more easily distinguish between similar emotions based on the evolving prototypes."
>
> **W3:** The study could examine how different types of noise augmentation affect model robustness. Currently, the mixup method is effective, but real-world noise may be more complex.
>
> **Reply:** As mentioned in lines 243-246 of the paper, "However, due to the continuous and time-series nature of EEG signals, traditional data augmentation techniques, such as cropping or scaling, are not applicable. Moreover, generating new signals using complex models like Generative Adversarial Networks (GANs) would dramatically increase the model’s complexity." Currently, there is limited research on noise augmentation for EEG signals without increasing model parameters. The noise augmentation method based on mixup introduced in this paper is innovative and has been validated through multiple datasets. We plan to explore and investigate additional data augmentation methods in future work.
>
> **Questions:**
>
> **Q1:** How does the model handle datasets like DEAP?
>
> **Reply:** The basic assumption of PLL is that only one of the multiple candidate labels is correct. However, DEAP labels are continuous signals in dimensions such as valence and arousal, which do not fit the assumption of PLL learning. When the DEAP dataset is used for classification, continuous labels on dimensions like valence are divided into binary or ternary categories, but these categories mainly reflect varying intensities along the valence dimension. They do not possess the inherent relationships seen between emotion categories, and therefore, are not suitable for our proposed semantic-based candidate label generation method. However, from a model perspective, our experiments in Appendix A.7 (lines 912-936) show that under the supervision of true labels, our proposed two modules, PG and NA, still provide performance gains. This suggests that under a fully supervised setting, our method may generalize to datasets like DEAP, particularly the noise augmentation module, which addresses the inherent low signal-to-noise ratio problem of EEG signals, a challenge also present in the DEAP dataset.
>
> **Q2:** What are the implications of using alternative embeddings?
>
> **Reply:** This paper is the first to introduce the use of the GloVe-based embedding dictionary to generate candidate labels, which is a clear innovation in the field. Especially in the context of real-world EEG emotion recognition tasks, there is currently no gold standard to simulate the candidate label generation process. Therefore, this exploration will contribute to the development of the affective computing community. We will explore other embeddings, such as using pre-trained BERT to generate word vectors, as part of our future work.
>
> **Q3:** How does the model address unseen emotions or emotions not mapped in the initial GloVe embedding setup?
>
> **Reply:** GloVe is a standard dictionary in the NLP field with a large vocabulary. We have added an introduction to GloVe in Appendix A.2.1 (lines 732-742). The version we used includes 840 billion tokens, which covers most emotional vocabulary, as emotional expressions are prevalent in large textual corpora. If we encounter unseen emotion words, they can be mapped to synonyms or semantically similar terms as substitutes. In the future, if emotion words do indeed fall outside the coverage of GloVe, we could further explore the approach of using methods like BERT to regenerate word embeddings for different words in the same space.

---

> ### Author Response · Authors · 2024-12-04
> **Thanks**
>
> Dear Reviewer 9xuQ,
>
> Thank you very much for your insightful and constructive feedback. We truly appreciate the time and effort you have dedicated to reviewing our work.
>
> We are particularly grateful for the open questions you raised, especially regarding the application of our method to other types of datasets, the exploration of alternative word vector generation techniques, and the adaptability to unseen emotions. These points are extremely thought-provoking and have provided us with valuable perspectives. We will certainly incorporate these considerations into our future work and continue to explore these areas in more depth.
>
> Once again, thank you for your valuable suggestions. We are confident that they will help guide the further improvement and extension of our research.
>
> Best regards,
>
> The authors

---

### Official Review · Reviewer_5Rnw · 2024-11-01

**Soundness:** 2
**Presentation:** 3
**Contribution:** 2
**Rating:** 3
**Confidence:** 5

**Summary:**

This paper aims to address challenges in semantic-based candidate label generation, signal-to-noise ratio (SNR) issues in EEG data using a novel prototype-based approach for decoding emotional relationships. The authors apply their method to three publicly available EEG-based datasets on emotion recognition, namely SEED, SEED-IV, SEED-V, and claimed the proposed method achieving the state-of-the-art performance over existing methods across these datasets.

**Strengths:**

The authors incorporate real-world ambiguous labeling simulation and partial label learning to address the challenge of annotation errors due to the complexity of emotions.

**Weaknesses:**

1. Semantic Similarity for Candidate Label Generation: The authors claim that the lack of semantic-based candidate label generation in existing partial label learning work for emotion recognition poses a challenge in the field.
However, this assumption is questionable, as it overlooks the psychological closeness of emotions. While GloVe embeddings were used to capture contextual word usage, these embeddings don’t reflect the psychological relationships between emotions. For example, in daily life, 'happy' and 'neutral' are often more easily confused than 'happy' and 'sad'. This difference has been correctly represented in Russell's model but not in the GloVe-based semantic panel. Specifically, as shown in Figure 3(b),  'happy'-'neutral' similarity is 0.5, while 'happy'-'sad' similarity is 0 in the Russell's circumplex model. However, in the semantic panel shown in Figure 3(a), 'happy'-'neutral' similarity is 0.21 while 'happy'-'sad' similarity is 0.64 (3x higher).
Furthermore, as shown in Figure 3, I think author made a mistake in the captions of Figures 3(a) and 3(b), where it should be 'inter-class similarity' instead of 'inter-class distance'. Since the authors mentioned 'cosine distance' only once in the manuscript (line 425) but mentioned 'cosine similarity' several times, and evidenced in the function 'Semantic_Distribution_similarity' in 'utils.py'(code line 69). Therefore, I strongly suggest that the authors clarify 'distance' in both the main text and figure captions as emotions are closer when their similarity is higher or distance is smaller.

2. Challenge of Low SNR in EEG: The authors claim to address the challenge of low SNR issue in EEG, as 'To mitigate the low SNR of EEG signals, inspired by the mixup method, we introduce a noise augmentation strategy, incorporating controllable noise to enhance model robustness'. However, this has already been done in CR[1] and PiCO[2] methods. The claim of 'advanced prototype-guided approach require a clearer clarification.

3. Marginal Performance Improvement: The authors claim that 'Experiments on three public datasets (SEED, SEED-IV, SEED-V) show that our approach achieves state-ofthe-art performance.' However, as shown in Tables 1, 2, & 3,the improvement are very marginal, in most cases less than 1%.

4. Evaluation Metric Choice: Why is F1 score chosen as the evaluation metric for the balanced datasets? Additionally, if F1 is used, why not include standard deviation (SD), as is done with accuracy? In 335-228, the authors claim 'The SEED-V dataset adopts a similar design to SEED-IV for cross-validation without the need for sample balancing, since five videos of different emotions always appear sequentially, leading to a 30:15 ratio of 'initial training set' to test set trials.' ...' However, as desribed in the original paper publishing the dataset [3], in SEED-V, each participant repeat session three times, each time contains 15 trials (3 per emotional category), this design ensures that the samples are balanced across emotions in both the training and test sets and no data leakage would occur. It can be also found in the official PyTorch Datasets link: https://torcheeg.readthedocs.io/en/latest/generated/torcheeg.datasets.SEEDVDataset.html#torcheeg.datasets.SEEDVDataset. Could authors provide evidence to support the claims of imbalanced dataset and data leakge in the train-test protocol proposed in [3] or in any dataset used in this study?


5. Comparision with Inappropriate Data Augmentation: Table 4 compares with an augmentation technique, namely additive pepper&salt noise, which is not recommened for EEG data augmentation as it can distort intrinsic EEG features [4, 5].

6. Code Ethics: The authors should properly acknowledge external sources to maintain transparency and integrity. For example, lines 28-309 in 'train_func.py' are directly copied from 'https://github.com/guangyizhangbci/PLL-Emotion-EEG/blob/main/train_func.py'. Similarly, most parts of the code in 'utils.py', even including comments, are identical to 'https://github.com/guangyizhangbci/PLL-Emotion-EEG/blob/main/utils.py'.



[1] Wu, Dong-Dong, Deng-Bao Wang, and Min-Ling Zhang. "Revisiting consistency regularization for deep partial label learning." International conference on machine learning. PMLR, 2022.

[2]Wang, Haobo, et al. "PiCO: Contrastive label disambiguation for partial label learning." International conference on learning representations. 2022.

[3] Liu, Wei, et al. "Comparing recognition performance and robustness of multimodal deep learning models for multimodal emotion recognition." IEEE Transactions on Cognitive and Developmental Systems 14.2 (2021): 715-729.

[4] Wang, Fang, et al. "Data augmentation for EEG-based emotion recognition with deep convolutional neural networks." MultiMedia Modeling: 24th International Conference, MMM 2018, Bangkok, Thailand, February 5-7, 2018, Proceedings, Part II 24. Springer International Publishing, 2018.

[5] Lashgari, Elnaz, Dehua Liang, and Uri Maoz. "Data augmentation for deep-learning-based electroencephalography." Journal of Neuroscience Methods 346 (2020): 108885.

**Questions:**

1. As mentioned above in the 'Weakness' section, authors should clarify the term 'inter-class distance' in both main text and figure captions.

2. The motivation for applying PLL methods for EEG-based emotion recognition needs further explanation. Specifically, why is PLL necessary in emotion recognition? If annotation error is the issue, how reliable is the labeling in the SEED-series datasets?

3. The SEED dataset includes only three emotions, namely 'happy', 'neutral', and 'sad', which are often not difficult to distinguish. Why not select datasets with a broader range of emotions where ambiguous labeling is more likely to occur?

---

> ### Author Response · Authors · 2024-12-02
> **Response to Reviewer 5Rnw**
>
> **Weaknesses:**
>
> **W1:** The claim that semantic-based candidate label generation is lacking in partial label learning (PLL) for emotion recognition is questionable. GloVe embeddings do not capture the psychological closeness between emotions, as shown by the intuitively unreasonable similarity scores among happy, neutral, and sad in Figure 3. The term “inter-class distance” in the figures should be replaced with “inter-class similarity,” and the authors should clarify the use of “distance” versus “similarity.”
>
> **Reply:** You mentioned that the Russell distribution's representation of similarity among emotions like happy-neutral-sad aligns more intuitively with human understanding. We acknowledge that this is one of the strengths of the Russell Distribution. However, it is not currently the gold standard for generating emotion candidate labels. The Russell Distribution also has some potential issues, such as the fact that the similarity between Neutral and all other emotions is fixed at 0.5, which is hard to justify. In contrast, the Semantic Distribution, derived from GloVe, captures semantic relationships that make the similarity between Neutral and other emotions more nuanced, reflecting the complexity of emotions. Furthermore, based on neural responses to emotions in specific brain regions, as mentioned in [1],
>
> > “the weights in the frontal and parietal lobes of positive and negative emotions are obviously higher than the neutral emotion.”
>
> This suggests that brain responses to positive and negative emotions are stronger than to neutral emotions in certain regions, implying that the EEG responses for emotional states may be closer to each other than to neutral states. This aligns with the semantic similarity between emotions in the GloVe-based distribution, supporting the rationale for using GloVe as an emotion dictionary for candidate label generation. We have added this argument in section 3.2.3 (lines 518-522).
>
> Additionally, regarding the errors in some of the captions in Figure 3, thank you for your detailed review. We have corrected the table headers in Figure 3 (a) and (b), changing “distance” to “similarity” and have added a description of the relationship between distance and similarity in section 3.2.3 (line 483).
>
> [1] Weibang Jiang, Yuting Lan, and Baoliang Lu. REmoNet: Reducing emotional label noise via multi-regularized self-supervision. In Proceedings of the 32nd ACM International Conference on Multimedia, pp. 2204–2213, 2024.
>
> **W2:** The noise augmentation method inspired by mixup is not novel, as similar methods have already been explored in CR and PiCO. The “prototype-guided approach” requires further clarification.
>
> **Reply:** PiCO uses two different data augmentation methods to obtain positive samples for contrastive learning, applying MoCo's contrastive learning paradigm, with data augmentation primarily based on geometric transformations. This approach does not apply to EEG. CR employs a similar consistency regularization method, which is conceptually similar, so we have referenced it in line 240. However, we are the first to adapt the mixup method to denoising tasks for EEG and have made two improvements for the PLL learning task:
>
> 1. We constrained the mixup ratio using equation (10) to ensure that the original sample retains more of its features. Through a hyperparameter search, we found that the best performance occurs when `\lambda'` is in the range of [0.8, 0.9]. This method ensures that the introduced noise remains controllable.
>
> 2. Unlike mixup, which interpolates the labels of the newly generated data, we keep the original labels unchanged in order to prevent the method from further confusing the labels in the PLL task. This approach aligns with the goal of noise augmentation rather than data augmentation. To further validate this, we have included a comparison with the standard mixup method under the same conditions in Appendix A.9 (lines 1006-1053), where we demonstrate that our method outperforms mixup in most cases. These improvements and experimental comparisons sufficiently demonstrate the innovation and contribution of our proposed noise-augmentation method.
>
> Moreover, to better clarify our proposed Prototype-Guided Module, we have rewritten section 2.3.1 (lines 190-235), providing a more detailed description of the method's implementation and advantages.

---

> ### Author Response · Authors · 2024-12-02
> **Response to Reviewer 5Rnw (Cont'd)**
>
> **W3:** The performance improvement of the proposed method is marginal, with less than 1% improvement in most cases. A clearer demonstration of significant performance gains is needed.
>
> **Reply:** The three datasets used in this study have different categories, subjects, and stimulus sources. There are significant differences in both data distribution and label distribution. Our method consistently outperforms on datasets with different distributions, and we have demonstrated through ablation experiments that our method achieves steady improvement over the DNPL baseline. Therefore, the experimental data are sufficient to validate the advantages of our method. In Appendix A.8 (lines 940-1003), we also include experiments conducted on another commonly used encoder, further demonstrating the superiority of our approach.
>
> **W4:** Why is F1 score chosen for balanced datasets, and why isn't standard deviation (SD) included as with accuracy? The authors claim a dataset imbalance in SEED-V, but this contradicts the dataset's design, which ensures balance. The authors should provide evidence to support the claim of data imbalance or leakage.
>
>
> **Reply:** In a balanced dataset, although the sample size for each class is similar, it does not mean that the model's performance is the same across all classes. Accuracy may mask imbalances in errors, whereas the F1 score provides a more comprehensive evaluation of the model's performance. Therefore, we have added two types of F1 scores for evaluation. We have adopted your suggestion and added the standard deviations of the two F1 scores in Tables 1-3.
>
> Regarding the dataset distribution imbalance, we describe in the paper the impact of the unordered labels in the SEED-IV dataset, not referring to the SEED-V dataset (e.g., in lines 346-347: "The SEED-IV dataset, with 24 trials per session, employs a unique approach due to the random order of trials."). Please refer to the SEED-IV dataset label distribution below, where each row represents the sequence of labels for the 24 trials in a session. Traditional methods typically split the training and test sets in each session with a 16:8 ratio. If only the first session is used, the test set would contain only labels 0 and 3. If three sessions are used, the test set would have a label distribution of 0:1:2:3 in a 5:3:1:3 ratio, leading to an imbalance between the training and test sets. Similar issues are discussed in [1]. To address this, we used a more balanced approach, dividing the trials into three folds for cross-validation, ensuring a more persuasive dataset partitioning and avoiding class imbalance.
>
> > [1,2,3,0,2,0,0,1,0,1,2,1,1,1,2,3,2,2,3,3,0,3,0,3],
> > [2,1,3,0,0,2,0,2,3,3,2,3,2,0,1,1,2,1,0,3,0,1,3,1],
> > [1,2,2,1,3,3,3,1,1,2,1,0,2,3,3,0,2,3,0,0,2,0,1,0]
>
> [1] Xuanhao Liu, Weibang Jiang, Weilong Zheng, Baoliang Lu. *Two-Stream Spectral-Temporal Denoising Network for End-to-End Robust EEG-Based Emotion Recognition* [C]// International Conference on Neural Information Processing. Singapore: Springer Nature Singapore, pp.186-197. 2023.
>
> **W5:** The comparison with an inappropriate data augmentation method (pepper&salt noise) is problematic, as it can distort EEG data features.
>
> **Reply:** Thank you for raising this point. We have removed this content from the main text and moved it to Appendix A.9 (lines 1006-1053). Additionally, in Appendix A.9, we provide a comparison with the original mixup method, further demonstrating the advantages of our proposed noise-augmentation method.
>
> **W6:** There are ethical concerns regarding code transparency. Sections of code in "train_func.py" and "utils.py" appear to be copied directly from an external source without proper acknowledgment.
>
> **Reply:** Our experimental code is based on the open-source code provided by the referenced work you mentioned, which is mainly used for reproducing the baseline results, not our innovations. We appreciate their contribution and have properly cited their work in the paper. Following your suggestion, we have also added an acknowledgment in the ReadMe file of the attached code. To avoid violating the anonymity of the review process, we have not directly mentioned the author's name or provided the link yet. After the code is officially released, we will update the acknowledgment accordingly.

---

> ### Author Response · Authors · 2024-12-02
> **Response to Reviewer 5Rnw (Cont'd)**
>
> **Questions:**
>
> **Q1:** As mentioned above in the 'Weakness' section, authors should clarify the term 'inter-class distance' in both main text and figure captions.
>
> **Reply:** As for Weakness 1, we have already optimized it at the corresponding position in Figure 3 of the manuscript.
>
> **Q2:** The motivation for applying PLL methods to EEG-based emotion recognition needs more explanation. Why is PLL necessary for emotion recognition, and how reliable is labeling in the SEED-series datasets?
>
> **Reply:** As mentioned in the Introduction (lines 44-49), emotions have a certain level of complexity, particularly in the evolution and interweaving of different emotions. Therefore, when classifying emotion categories, labeling errors may occur due to these factors. The SEED series datasets are already widely recognized and commonly used high-quality datasets. Nevertheless, It is pointed out in [1] that the video-level labeling method in SEED series datasets might introduce noisy labels, as follows:
>
> > "Despite the promising performance achieved by numerous existing methods, several challenges persist. Firstly, there is the challenge of emotional label noise, stemming from the assumption that emotions remain consistently evoked and stable throughout the entirety of video observation."
>
> This highlights that noise in labels is one of the inevitable challenges in the field of EEG-based emotion recognition. While our method is based on the SEED series datasets, the relevant noisy labels are simulated using two different candidate label generation methods. The goal is to address the disambiguation problem caused by these candidate labels, rather than to solve the noise issue inherent in the SEED datasets themselves.
>
> [1] Weibang Jiang, Yuting Lan, and Bao-Liang Lu. *REmoNet: Reducing emotional label noise via multi-regularized self-supervision*. In Proceedings of the 32nd ACM International Conference on Multimedia, pp. 2204–2213, 2024.
>
> **Q3:** The SEED dataset only includes three emotions, which are easy to distinguish. Why not select a dataset with a broader range of emotions where ambiguous labeling is more likely to occur?
>
> **Reply:** We conducted experiments on three different SEED series datasets, with the SEED-V dataset containing the largest number of emotion categories (five), including three distinct negative emotion categories. We performed comparative experiments under two different experimental distributions across these three datasets, and the abundant experimental results are sufficient to demonstrate the advantages of our method. Although there are a few datasets with more emotion categories, such as MPED [1] and FACED [2], we recognize that multi-class emotion recognition itself is an important challenge, and current works in this area still show relatively low accuracy. Evaluations on MPED [3-5] and FACED [5] have reported that the accuracy is generally below 40%, making further evaluation of PLL susceptible to the impact of the encoder itself. Therefore, we have chosen the SEED series datasets with higher accuracy to validate the effectiveness of the proposed PLL method.
>
> [1] Tengfei Song, Wenming Zheng, Cheng Lu, Yuan Zong, Xilei Zhang, and Zhen Cui. MPED: A multi-modal physiological emotion database for discrete emotion recognition[J]. IEEE Access, 7: 12177-12191. 2019
>
> [2] Jingjing Chen, Xiaobin Wang, Chen Huang, Xin Hu, Xinke Shen, and Dan Zhang. A large finer-grained affective computing EEG dataset[J]. Scientific Data, 10(1): 740. 2023
>
> [3] Tengfei Song, Suyuan Liu, Wenming Zheng, Yuan Zong, and Zhen Cui. 2020. Instance-adaptive graph for EEG emotion recognition. In Proceedings of the AAAI Conference on Artificial Intelligence. pp.2701–2708.
>
> [4] Yang Li, Ji Chen, Fu Li, Boxun Fu, Hao Wu, Youshuo Ji, Yijin Zhou, Yi Niu, Guangming Shi, and Wenming Zheng. 2022. GMSS: Graph-based multi-task self-supervised learning for EEG emotion recognition. IEEE Transactions on Affective Computing. 14, 2512–2525. 2022.
>
> [5] Wei Li, Lingmin Fan, Shitong Shao, and Aiguo Song. Generalized contrastive partial label learning for cross-subject EEG-based emotion recognition. IEEE Transactions on Instrumentation and Measurement, 73, 2024.

---

> > ### Comment · Reviewer_5Rnw · 2024-12-03
> >
> > I would like to thank all the authors for their efforts in revising the manuscript and addressing the comments. I acknowledge the response and have adjusted my evaluation accordingly.

---

> > > ### Author Response · Authors · 2024-12-03
> > > **Thanks**
> > >
> > > Dear Reviewer 5Rnw,
> > >
> > > Thank you for your thorough review and the valuable suggestions you provided. We particularly appreciate the detailed comparison and analysis you made between the Russell Distribution and the Semantic Distribution proposed in our paper. Your insights into the strengths and weaknesses of both approaches have been extremely helpful and will motivate us to further optimize the related methods.
> > >
> > > We are also grateful for the positive adjustment in your score. Your support has strengthened our confidence in advancing our research in this area.
> > >
> > > Once again, thank you for your time, effort, and constructive feedback. It has been invaluable to our work.
> > >
> > > Best Regards,
> > >
> > > The authors

---

### Official Review · Reviewer_uAJS · 2024-11-01

**Soundness:** 2
**Presentation:** 2
**Contribution:** 2
**Rating:** 6
**Confidence:** 4

**Summary:**

The paper proposed a new method for EEG-based emotion recognition. The method used the semantic relationships between emotions and some prototype, and used a noise augmentation strategy to enhance model robustness. The experiments were done on three public EEG datasets. The paper was well organized.

**Strengths:**

The work introduced a new aspect to the EEG emotion recognition field, partial label learning. The target problem was clear, the emotion labels might not be accurate and EEG has low SNR.

**Weaknesses:**

Although the proposed method might provide a new aspect to the problem. But it seemed that the authors might not be fully familiar to EEG analysis. The study was incremental and the contribution was limited. The proposed method was the combination of existing methods in other domains and be modified to EEG analysis. And it was not well modified. For example, the model structure did not fully consider EEG temporal-spatial-spectral information.

**Questions:**

1. The performance improvement of the proposed method seemed small on the three datasets compared to the baselines. Was the improvement significant enough to draw a conclusion that the proposed method outperformed others?
2. In the experiments, the baselines were not the classic baselines that were often used in EEG analysis. It seems that the work simply took some machine learning methods in other domains and applied them to the EEG domain. Has the authors compared the model performance with other baselines on the same datasets in the literature?
3. How many prototypes are needed to update in the model? Was it possible to use more prototypes and how did the number of prototypes influence the model performance?
4. There were many writing typos in the paper, the authors should carefully check the English writing.

---

> ### Author Response · Authors · 2024-12-02
> **Response to Reviewer uAJS**
>
> ### **Weaknesses:**
>
> **W1:**  The authors may not be fully familiar with EEG analysis. The study appears incremental, with limited contribution, as the proposed method combines existing approaches from other domains without fully adapting them to EEG analysis, particularly ignoring EEG's temporal-spatial-spectral features.
>
> **Reply:**  I understand that the temporal-spatial-spectral feature extraction you mentioned is important for constructing robust EEG emotion recognition encoders. However, our goal is to address the PLL problem in the context of EEG emotion recognition, not to explore a more robust EEG feature extraction method. Our model architecture is based on [1][2], where the encoder leverages CNNs to extract high-dimensional features from frequency-domain feature DE. To further demonstrate the performance advantages of our method on other common encoders, we have added a comparison of results on another widely used encoder, MLP, in Appendix A.8 (lines 940-1003). In particular, the five citations listed in lines 943-944 show that cross-subject tasks and semi-supervised learning tasks related to EEG-based emotion recognition are often evaluated using MLP. We compare different baselines, and our method also shows superior performance on this encoder. The experiments on these two model architectures are sufficient to validate the effectiveness of our method.
>
>
> ### **Questions:**
>
>  **Q1:**  The performance improvement of the proposed method seems small compared to the baselines. Was the improvement significant enough to draw a conclusion that the proposed method outperformed others?
>
>  **Reply:**  The three datasets used in this paper involve different categories, subjects, and stimulus sources, leading to substantial differences in both data and label distributions. Our method consistently outperforms others on datasets with different distributions, and ablation experiments demonstrate a steady improvement over the baseline DNPL. Thus, the experimental data provided are sufficient to prove the advantages of our method. Additionally, the experiments in Appendix A.8, which compare the performance of our method on another commonly used encoder MLP, further confirm the superiority of our approach.
>
> **Q2:**  The baselines used in the experiments are not typical for EEG analysis. Has the method been compared to more classic EEG baselines used in the literature?
>
> **Reply:**  Research on PLL in the context of EEG emotion recognition is very limited, and the baseline methods on related datasets are usually developed under fully supervised conditions, rather than addressing the PLL problem. Therefore, comparisons with these works would not be fair. As far as we know, there are only two existing studies on PLL in EEG emotion recognition. [1] introduces the PLL problem to EEG emotion recognition, and [3] proposes a model that simultaneously addresses PLL and cross-subject issues but has not been open-sourced, making it difficult to perform a fair comparison under the same experimental settings. Notably, the baselines in these two works are classical PLL learning methods from computer vision tasks. These methods have been fairly compared in our setup. Our work is a novel PLL learning method specifically tailored for EEG emotion recognition, and we provide reproducible source code. Our method sets a new baseline for PLL problems in this domain.
>
>  **Q3:**  How many prototypes are needed to update in the model? Was it possible to use more prototypes and how did the number of prototypes influence the model performance?
>
>  **Reply:**  The approach you mentioned is worth exploring further. In this paper, we use only a single prototype, similar to methods like PiCO. Intuitively, introducing multiple prototypes and leveraging a mixture-of-experts approach could potentially enhance the model's performance. However, the construction of prototypes relies on a moving average method without introducing new parameters, so adding more prototypes might not significantly help. Nevertheless, following this line of thought, we will attempt further exploration in future work.
>
>  **Q4:**  There are numerous typos throughout the paper that need to be corrected.
>
>  **Reply:**  Thank you for your careful review of the paper's details. We have thoroughly re-examined the entire paper and made detailed optimizations.
>
> [1] Guangyi Zhang and Ali Etemad. Partial label learning for emotion recognition from EEG. arXiv preprint arXiv:2302.13170, 2023.
> [2] Guangyi Zhang, Vandad Davoodnia, and Ali Etemad. PARSE: Pairwise alignment of representations in semi-supervised EEG learning for emotion recognition. IEEE Transactions on Affective Computing, 13(4):2185–2200, 2022a.
> [3] Wei Li, Lingmin Fan, Shitong Shao, and Aiguo Song. Generalized contrastive partial label learning for cross-subject EEG-based emotion recognition. IEEE Transactions on Instrumentation and Measurement, 73, 2024.

---

> > ### Comment · Reviewer_uAJS · 2024-12-03
> >
> > Thank you for your detailed response. I have raised my scores.

---

> > > ### Author Response · Authors · 2024-12-03
> > > **Thanks**
> > >
> > > Dear Reviewer uAJS,
> > >
> > >
> > > Thank you very much for your thoughtful feedback and constructive comments. We are grateful for the positive evaluation of the revised manuscript, and we are thrilled that your support has contributed to the improvement in the score. Your encouragement has significantly strengthened our confidence in further exploring the relevant research area.
> > >
> > >
> > > Your detailed review and insightful suggestions played a critical role in enhancing the quality of our paper. We carefully implemented your recommendations, which have not only improved the manuscript but also provided us with valuable guidance for our future work.
> > >
> > >
> > > Once again, thank you for your time, effort, and invaluable support.
> > >
> > >
> > > Best regards,
> > >
> > > The authors

---

### Official Review · Reviewer_vK4J · 2024-11-02

**Soundness:** 2
**Presentation:** 2
**Contribution:** 2
**Rating:** 6
**Confidence:** 3

**Summary:**

This paper introduces a novel EEG-based emotion recognition model, PGNA-PL, which utilizes prototype-guided disambiguation and noise augmentation to improve accuracy in partial label learning (PLL) settings. By leveraging GloVe embeddings to generate semantically informed candidate labels, the approach addresses label ambiguity and low SNR in EEG signals, achieving state-of-the-art results across multiple datasets.

**Strengths:**

1. The setting of partial label learning for EEG-based emotion recognition is relatively new and interesting.
2. Experiments on three challenging EEG datasets (SEED, SEED-IV, and SEED-V) reveal the approach’s effectiveness in both accuracy and F1 measures.
3. The code is provided and implementation details are comprehensive.

**Weaknesses:**

1. The emotional semantic labels derived via GloVe are innovative, but it's unclear how these candidate labels compare to using the original dataset labels directly. A comparative analysis between semantic-based and default labels would help clarify the added value of semantic embedding, especially as GloVe vectors may not always capture the full emotional nuance specific to EEG-based emotion recognition.
2. Although the noise augmentation module is inspired by mixup, the connection to addressing low SNR remains speculative without clear comparative evidence. The paper would benefit from directly comparing PGNA-PL’s noise augmentation with a simple mixup approach to demonstrate its unique effectiveness. A straightforward ablation study excluding the noise augmentation does not sufficiently confirm its necessity; a specific baseline comparison with mixup is essential to establish its impact and necessity in the model's performance.
3. While the study describes its method as a partial label learning (PLL) approach, it seems to depend heavily on fully annotated labels from the datasets to generate candidate labels. Traditional PLL assumes that a data sample's true label is unknown among several candidates, yet here, the candidates seem to be predetermined using GloVe embeddings for emotion categories. This difference in the PLL approach could merit further discussion, particularly regarding the rationale for using the same candidate labels across emotion categories and the method's applicability in true PLL settings.
4. The method's performance would be more convincing if compared with a fully supervised version of the proposed model, where true labels are directly utilized without PLL. This comparison would reveal the performance trade-offs or benefits PGNA-PL offers in balancing between label ambiguity and robust emotion classification.

**Questions:**

1. The tables refer to Semantic Distribution and Russell Distribution as label generation methods, but further explanation is needed. Could the authors clarify the conceptual and practical differences between these two approaches in the context of EEG-based emotion recognition? Additionally, it would help to explain how each distribution affects the model's performance across different emotion categories, given that these distributions likely represent different assumptions on emotion relationships.
2. What is the value of \beta_p in Equation 9?

---

> ### Author Response · Authors · 2024-12-02
> **Response to Reviewer vK4J**
>
> ### **Weaknesses:**
>
> **W1:** The comparison between GloVe-based semantic labels and original dataset labels is unclear. A direct comparison would highlight the value of semantic embedding in EEG-based emotion recognition.
>
> **Reply:**  Thank you for your suggestion. We leverage the rich semantic information provided by GloVe to capture the relationships between different emotions, simulating the situation in which errors occur during the labeling process due to confusion between emotions. In this case, it is reasonable that the performance of the same model is lower than that of fully supervised learning. We have added the performance of PGNA-PL under fully supervised conditions (using the original dataset labels) in Table 6 (line 930). Based on the data provided in Table 6, the comparison between PGNA-PL's performance in Semantic Distribution and fully supervised learning is shown in the table below. It can be observed that the Semantic Distribution performance is very close to the fully supervised performance, indicating that our proposed candidate label generation method is reasonable. However, we did not include this discussion in the paper because labels under mislabeling conditions do not have an "advantage" over fully supervised labels. Nevertheless, inspired by your suggestion, we have added a comparison between PGNA-PL’s performance under fully supervised conditions and the backbone (encoder + decoder) under fully supervised conditions in Table 6. We find that our proposed model also outperforms the backbone model under fully supervised conditions. This additional experiment will help demonstrate the generalization capability of the PLL model we proposed.
>
> **Table. Comparison of performance metrics of PGNA-PL under Semantic Distribution and full supervision on the SEED, SEED-IV, and SEED-V datasets**
> | Method                                  | SEED Accuracy       | SEED Macro F1       | SEED Micro F1       | SEED-IV Accuracy    | SEED-IV Macro F1    | SEED-IV Micro F1    | SEED-V Accuracy     | SEED-V Macro F1     | SEED-V Micro F1     |
> |-----------------------------------------|---------------------|---------------------|---------------------|---------------------|---------------------|---------------------|---------------------|---------------------|---------------------|
> | PGNA-PL (Semantic Distribution)         | 79.69 ($\pm$11.86)  | 78.11 ($\pm$13.11)  | 79.31 ($\pm$12.07)  | 68.31 ($\pm$12.55)  | 67.02 ($\pm$12.69)  | 67.86 ($\pm$12.11)  | 60.11 ($\pm$16.42)  | 58.02 ($\pm$16.32)  | 60.4 ($\pm$15.24)   |
> | **PGNA-PL (full supervision)**          | **80.11 ($\pm$12.02)** | **78.47 ($\pm$13.4)** | **79.7 ($\pm$12.25)** | **68.8 ($\pm$12.08)** | **67.49 ($\pm$12.29)** | **68.36 ($\pm$11.75)** | **61.02 ($\pm$16.14)** | **59.4 ($\pm$16.04)** | **61.62 ($\pm$14.77)** |
>
>
> **W2:** The noise augmentation method's connection to low SNR is speculative without a direct comparison to mixup. A comparison with mixup is needed to confirm the method's effectiveness and necessity.
>
> **Reply:**  We have followed your suggestion and included a comparison with the original mixup method in Appendix A.9. It utilizes the same hyperparameters as our method, such as $\beta_p$ and $\sigma$ in Equations 9-10, to ensure a fair comparison. On different datasets, PGNA-PL achieves superior results in most experiments. The original mixup method may further increase label confusion, which poses a potential risk for label noise learning and could explain its poorer performance.
>
> **W3:** The method seems to rely on fully annotated labels for candidate generation, which differs from traditional PLL. The use of GloVe embeddings for candidate labels should be discussed in the context of true PLL settings.
>
> **Reply:**  Our hypothesis is essentially consistent with traditional PLL, which corresponds to the scenario you mentioned, where "the true label is unknown among multiple candidate labels." However, not all emotion categories use the same candidate labels. As stated in lines 147-148: "Equations (2) and (3) are applied sequentially to the emotion labels of each sample. As a result, samples with the same true label may be assigned different candidate labels." We calculate inter-class relationships using semantic information provided by GloVe to simulate the generation probability of candidate labels. This is consistent with the basic framework of traditional PLL algorithms, which use a uniform distribution to simulate the generation probability of candidate labels. However, our proposed method is more suitable for emotion recognition, which is the unique feature of our approach.
>
> **W4:** A comparison with a fully supervised model using true labels is needed to demonstrate the trade-offs or benefits of PGNA-PL in emotion classification.
>
> **Reply:**  The relevant data has been provided in response to question 1.

---

> ### Author Response · Authors · 2024-12-02
> **Response to Reviewer vK4J (Cont'd)**
>
> ### **Questions:**
>
>  **Q1:**  The difference between Semantic Distribution and Russell Distribution is unclear. Could the authors explain their conceptual and practical differences in EEG-based emotion recognition and their impact on model performance across emotion categories?
>
> **Reply:**   We have added an explanation of the Russell Distribution in Appendix A 2.2 (lines 744-758). The Semantic Distribution and Russell Distribution are two different methods for generating candidate labels, both of which aim to use a predefined emotional distance relationship to simulate the candidate label generation process under mislabeling conditions. The two methods differ in how they define the relationships between emotion categories. The former is based on the GloVe dictionary (lines 732-742), which uses a language model trained on a large corpus of text to generate emotion vectors for different basic vocabulary items. The similarity between these vectors can represent the distance relationship between different words. The latter uses Russell’s circumplex model, which maps emotions on a circular layout based on arousal and valence, with each emotion defined by its polar coordinates (radius and angle). To generate candidate labels, it calculates the polar coordinates for each emotion and computes a normalized similarity score based on the cosine of the angular differences and their Euclidean distances on the circle.
>
>  **Q2:**  What is the value of $\beta_p$ in Equation 9?
>
> **Reply:**  The value range is [0.05-0.4]. Within this range, we conducted a grid search with intervals of 0.05 to identify the optimal $\beta_p$ across different datasets and settings. We have added this information in Appendix A.5 (lines 815-816).

---

> > ### Comment · Reviewer_vK4J · 2024-12-03
> >
> > I have carefully reviewed all the comments and responses. While I still have specific concerns regarding the rationale behind using GloVe embeddings and the novelty of noise augmentation, the rebuttal has addressed several of my other concerns. As a result, I have adjusted my score accordingly.

---

> > > ### Author Response · Authors · 2024-12-04
> > > **Thank You and Further Clarifications Regarding Reviewer vK4J's Concerns**
> > >
> > > Dear Reviewer vK4J,
> > >
> > > Thank you for your careful review of our manuscript and for taking the time to consider our rebuttal. We greatly appreciate your thoughtful comments, which have been invaluable in helping us refine our work.
> > >
> > > We are pleased to hear that our responses have addressed several of your concerns and that you have adjusted your score accordingly. However, we understand that you still have specific concerns regarding the rationale behind using GloVe vectors and the novelty of noise augmentation. We would like to take this opportunity to further clarify these points:
> > >
> > > ### 1. **Regarding the rationale behind using GloVe vectors**:
> > >
> > > To simulate the mislabeling process caused by label confusion, traditional methods apply uniform distribution sampling to determine the probability of an incorrect label becoming a candidate label. As you mentioned in W3, the candidate labels generated in this case do not rely on the original annotations. However, in emotion recognition, the complexity and interweaving of emotions often lead to label ambiguity (like the example in lines 46-49), which is influenced by two main factors:
> > >
> > > - The inherent **semantic relationships** between different emotions.
> > > - The **personalized influence** of annotators and participants on emotional perception and cognition.
> > >
> > > The first factor is considered a consensus across a wider population, representing relatively stable relationships. The second factor depends on individual differences in emotional perception and experience, making it more personalized. Because traditional uniform distribution methods struggle to account for these factors, they are less suitable for simulating mislabeling in emotional categories.
> > >
> > > Our approach focuses on the first factor—the inherent semantic relationships between emotions. We selected GloVe vectors (detailed in Appendix A.2.1, lines 732-742) because they are based on large-scale corpora, capturing the relationships between words, including emotional expressions. Since emotional words frequently co-occur in large corpora, we believe that GloVe vectors effectively simulate the inherent relationships between emotion words. In the initial stages of applying PLL evaluation methods and algorithms to emotion recognition, this approach provides a reasonable optimization direction for identifying candidate labels.
> > >
> > > As suggested by Reviewer 9xuQ, we also recognize the potential of using Transformer-based models like BERT to generate word vectors, which could address challenges such as generating vectors for unseen emotional words.
> > >
> > > Regarding the second aspect, which is ambiguity arising from personalized human understanding: How to encode personalized factors such as personality traits (e.g., extroversion, introversion) and use them to guide candidate label generation and the optimization of the PLL algorithm could become an interesting new problem. We believe this is an exciting direction for future research, and we are eager to explore how such personalized factors could be incorporated into our model to improve its accuracy in real-world scenarios.

---

> > > ### Author Response · Authors · 2024-12-04
> > > **Thank You and Further Clarifications Regarding Reviewer vK4J's Concerns (Cont'd)**
> > >
> > > ### 2. **Regarding the novelty of noise augmentation**:
> > >
> > > When we introduced the mixup method, we made an assumption, As Reviewer 5NVi appreciated a sentence in our paper in detailed notes 22: "Considering that the differences in EEG signal distributions between different emotional categories are smaller than those between EEG signals and other types of noise,..." this assumption is novel in EEG-related tasks. Our new noise augmentation experiments (see Appendix 9, lines 1006-1053) demonstrate that the mixup method is particularly suitable for EEG emotion recognition.
> > >
> > > However, unlike the original mixup method, which generates new data samples for augmentation, our approach focuses on adding noise to the original data. This incurs very low computational cost and does not increase the model’s parameter size.
> > >
> > > The original mixup method uses the λ value (from Equation 9, lines 257-258) as a balancing parameter in Equation 11 (lines 266-267) and requires label interpolation to generate new samples and labels for data augmentation. To better adapt the method to the PLL and EEG noise augmentation tasks, we made two key adjustments:
> > >
> > > - **First**, we introduced Equation 10 (lines 263) to account for the possibility that the true labels of different samples may not align. Excessively mixing features from different categories could disrupt the relevant category features. To address this, we used a larger σ value to preserve more of the original data’s features. Through grid search, we found that values in the range [0.8, 0.9] were most effective.
> > >
> > > - **Second**, in the PLL setting, where the true label is unknown among multiple candidates, label interpolation could exacerbate label ambiguity. We optimized this by directly retaining the original sample's label, avoiding further confusion of labels.
> > >
> > > We have incorporated your suggestion from W2 and added a comparison with the original mixup method in Appendix 9 (lines 1006-1053). In this comparison, we applied our proposed Equation 10 to mixup, with consistent hyperparameters. In most cases, our method outperforms the original, and the experiments validate the effectiveness of the improvement mentioned in the second point.
> > >
> > > We believe these modifications align the method with the goals of noise augmentation while preventing further label ambiguity in the PLL setting.
> > >
> > > Once again, thank you for your detailed review and valuable feedback. In particular, the concerns you raised have prompted us to think more deeply about how to clarify the innovation of these methods in our paper. We will incorporate these reflections into the revised manuscript.
> > >
> > > We apologize for the delayed submission of our rebuttal, as this may prevent further communication at this stage. However, after the anonymous review period ends, we would be happy to engage in further discussions with you regarding these points.
> > >
> > > Best regards,
> > >
> > > The authors

---

### Author Response · Authors · 2024-12-02
**Global Response to All Reviewers**

We sincerely thank the reviewers for their thorough evaluation of our paper and for providing many insightful comments. We apologize for the delay in submitting our rebuttal. In response, we have conducted additional experiments and carefully considered each of the suggestions, making optimizations to address the related issues in the revised version. The core improvements are as follows:

**1. Novelty:**
We are grateful to reviewers **vK4J**, **9xuQ**, and **5NVi** for their recognition of the novelty of our proposed methods. We have further refined the abstract (lines 15–23) to highlight the three key innovations more clearly, aiming to help readers better understand our contributions. These innovations include:
(i) A candidate label generation method for evaluating PLL algorithms.
(ii) A prototype-guided approach to help the model learn inter-class distance relationships for improved disambiguation.
(iii) A noise augmentation method designed to address the inherent low signal-to-noise ratio in EEG.
These methods provide new evaluation strategies and algorithms for emotion recognition in EEG-based PLL scenarios from various perspectives.

**2. Experimental Validation:**
Following the suggestions of reviewers **5Rnw** and **exz1**, we have included the standard deviations of the two F1 scores in Tables 1–3 and updated the related code accordingly. We also appreciate the feedback from reviewers **vK4J**, **uAJS**, and **exz1**, which led to the addition of experiments in Appendix A.7, A.8, and A.9, including:
(i) Fully supervised experiments using original labels.
(ii) Experiments with an alternative encoder (MLP).
(iii) Comparisons between our noise augmentation method and the original mixup method.
These additional experiments further strengthen the evidence supporting the effectiveness of our approach.

**3. Clarity of Expression:**
We especially thank reviewer **5NVi** for providing numerous detailed suggestions on improving clarity. We have carefully addressed each point and corrected all identified typos. In particular, we have rewritten and optimized the Abstract, Methods, and Related Work sections (the last of which has been moved to Appendix A.1). We also appreciate the clarity-related feedback from reviewers **uAJS**, **5Rnw**, and **exz1**.

**4. Open Questions:**
Additionally, we are grateful to reviewers **uAJS**, **9xuQ**, and **exz1** for raising open-ended questions, which have inspired us for future directions of our work.

Once again, we deeply appreciate the reviewers' valuable suggestions! To avoid making our response too lengthy, we have condensed the reviewers' suggestions into short phrases for clarity in our replies. We provide detailed responses to each of the reviewers' comments below and welcome further feedback on our revised paper.

---

### Meta-Review · Area_Chair_HM3h · 2024-12-20

**Metareview:**

The paper introduces an EEG-based emotion recognition framework  combining prototype-guided self-distillation and noise augmentation to enhance Partial Label Learning. The strengths of the paper include the important and timely problem statement, the interesting use of leveraging GloVe embeddings for label disambiguation, strong results, and details on implementation and code. However, the paper has a number of limitations. The novelty of the proposed method seems to be incremental, with a combination of a few well-known and simple steps (e.g., mixup and semantic embeddings) without major modifications tailored for EEG. Another issue in the paper is the concerns raised regarding the GloVe embeddings and the fact that they may not always align with psychological relationships between emotions, as well as its limited adaptability to dynamic datasets or unseen emotions. Another shortcoming is the use of datasets with limited emotion categories. Based on these and following the reviewer/AC discussions, I believe the weaknesses of the paper in its current form are enough to place the paper a bit below the acceptance threshold.

**Additional Comments On Reviewer Discussion:**

The paper has received scores of 3, 5, 5, 5, 6, 6. While the reviewers acknowledge the interesting concept of the paper, the general sentiment was that the paper requires further changes before being published. For instance, Reviewer 5Rnw raised concerns about the similarity between 'happy' and 'neutral' being reported as 0.21, while the similarity between 'happy' and 'sad' being reported to be 0.64 (3x higher), with which Reviewer 9xuQ agreed. Reviewers 5NVi and exz1 raised concerns regarding the novelty of the work. Reviewers uAJS and vK4J who both gave the only scores of 6 did not participate in the discussion phase with the other reviewers and AC to support their assessments. In the end, I would like to thank Reviewer 5NVi for providing such detailed and constructive feedback.

---

### Decision · Program_Chairs · 2025-01-22

Reject